



# JOANNE : Joint dropsonde Observations of the Atmosphere in tropical North atlaNtic meso-scale Environments

Geet George[1], Bjorn Stevens[1], Sandrine Bony[2], Robert Pincus[3,4], Chris Fairall[4], Hauke Schulz[1], Tobias Kölling[1], Quinn T. Kalen[5,*], Marcus Klingebiel[6,*], Heike Konow[7,*], Ashley Lundry[5,*], Marc Prange[8,9,*], and Jule Radtke[8,9,*]

[1]Max Planck Institute for Meteorology, Hamburg, Germany
[2]LMD/IPSL, Sorbonne University, CNRS, Paris, France
[3]Cooperative Institute for Research in Environmental Sciences, University of Colorado, Boulder, Colorado, USA
[4]NOAA Physical Sciences Laboratory, Boulder, Colorado, USA
[5]NOAA Aircraft Operation Center, Lakeland, Florida, USA
[6]Institute for Meteorology, University Leipzig, Leipzig, Germany
[7]Meteorological Institute, Universität Hamburg, Hamburg, Germany
[8]Meteorological Institute, Center for Earth System Research and Sustainability, Universität Hamburg, Hamburg, Germany
[9]International Max Planck Research School on Earth System Modelling, Max Planck Institute for Meteorology, Hamburg, Germany
[*]These authors contributed equally to the work

**Correspondence:** Geet George (geet.george@mpimet.mpg.de)

**Abstract.** As part of the EUREC⁴A field campaign which took place over the tropical North Atlantic during January-February 2020, 1216 dropsondes from the HALO and WP-3D aircraft were deployed through 26 flights to characterize the thermodynamic and dynamic environment of clouds in the trade-wind regions. We present *JOANNE* (*Joint dropsonde Observations of the Atmosphere in tropical North atlaNtic meso-scale Environments*), the dataset that contains these dropsonde measurements and the products derived from them. Along with the raw measurement profiles and basic post-processing of pressure, temperature, relative humidity and horizontal winds, the dataset also includes a homogenized and gridded data set with 10 m vertical spacing. The gridded data are used as a basis for deriving diagnostics of the area-averaged meso-scale circulation properties such as divergence, vorticity, vertical velocity and gradient terms, making use of sondes dropped at regular intervals along a circular flight path. 85 such circles, ∼222 km in diameter, were flown during EUREC⁴A. We describe the sampling strategy for dropsonde measurements during EUREC⁴A, the quality control for the data, the methods of estimation of additional products from the measurements and the different post-processed levels of the dataset. The dataset is publicly available (https://doi.org/10.25326/221) as is the software used to create it (https://doi.org/10.5281/zenodo.4746313).

## 1 Introduction

"EUREKA! This is what I want to study for the rest of my life."

In an exclamation of serendipitous prescience Joanne Simpson is reported to have said these words upon learning about the possibility of studying trade-wind cumulus clouds through airborne measurements (Fleming, 2020). Her subsequent research





proved foundational for tropical meteorology. Some seven decades later, the 2020 *ElUcidating the RolE of Cloud-Circulation Coupling in ClimAte* (EUREC[4]A) field campaign, unwittingly expressed her exclamation of enthusiasm in finding purpose on the same topic.

The EUREC[4]A field-campaign took place in January-February, 2020 and comprised measurements from many platforms. It adopted Barbados as its base of operations and focused its measurements in an area extending eastward of the Barbados Cloud Observatory (BCO; Stevens et al., 2016). EUREC[4]A's initial scientific motivation, its subsequent evolution, and the final execution are described in Bony et al. (2017) and Stevens et al. (2021). As these papers emphasise, a central element of EUREC[4]A was the airborne release of dropsondes to characterise the mesoscale meteorological environment of cloud fields

in the trades. The dropsondes were mostly deployed to enable accurate estimates of the mean vertical motion field, using an approach inspired by Lenschow et al. (1999, 2007) and adapted to dropsondes by Bony and Stevens (2019). Beyond estimating meso-scale vertical motion, the dropsondes were also aimed at characterising the thermodynamic structure in this region. In the stratified atmosphere of the trades, the dropsondes can resolve strong vertical gradients in temperature and moisture over short vertical distances, which are difficult to measure through remote-sensing (Stevens et al., 2017). The dropsondes are

thus essential in characterising the atmospheric environment within which many complementary measurements took place during EUREC[4]A. The purpose of this manuscript is to describe the resultant dropsonde dataset, which we call the *Joint dropsonde Observations of the Atmosphere in tropical North atlaNtic meso-scale Environments*, or *JOANNE*, in honour of Joanne Simpson's seminal contributions to our field of research.

*JOANNE* comprises five levels of data products, with each successive level encompassing a greater degree of synthesis and

post-processing. The basic measurements that go into the *JOANNE* data products, and how they were made, are discussed in Section 2. Quality control (QC) on the data are explained in Section 3, and evidence for a possible dry bias is presented in Section 4. The different levels of data products, and how they were constructed, are described in Section 5 and Section 6 concludes with a brief summary.

## 2    Sampling and Measurements

### 40    2.1    Instrument and Sensors

*JOANNE* is based entirely on data collected by Vaisala's RD-41 dropsondes (hereafter also 'sondes'; Vaisala, 2020a). A drop-sonde is similar to a radiosonde, with the exception that it is designed to be launched out of airborne platforms and sinks down through the atmosphere to the surface while making measurements. Each sonde has a cylindrical, cardboard casing that houses within it the measurement sensors, a GPS receiver, a battery and a signal transmitter for communicating with the airborne

receiving station. The casing is attached to a parachute that is designed to align the sonde properly for measurements and to reduce the fall speed.

The sondes carry three sensors - one each for measuring pressure ($p$), temperature ($T$) and relative humidity ($RH$), together referred to as the PTU sensors and with a sampling frequency of $2\,\mathrm{Hz}$. The GPS receiver allows the position of the dropsonde to be tracked, from which ambient winds are estimated at a sampling frequency of $4\,\mathrm{Hz}$. The sensors included in the sondes





**Table 1.** Details about sensors used in the RD-41 and RS-41 sondes are provided. Repeatability is the standard deviation of differences in twin soundings. The values for the sensors are obtained from Vaisala (2020a) and values for wind measurements estimated from GPS are obtained from Vaisala (2020b). All numbers are provided in terms of absolute units, correspondingly in the first column.

| Sensor / Measurements (Units) | Type | Range | Least Count | Repeatability |
|---|---|---|---|---|
| Pressure (hPa) | Silicon capacitor | surface pressure to 3 | 0.01 | 0.4 |
| Temperature (°C) | Platinum resistor | -90 to +60 | 0.01 | 0.1 |
| Relative Humidity (%) | Thin-film capacitor | 0 to 100 | 0.1 | 2 |
| Wind speed ($\mathrm{m\,s^{-1}}$) | estimated from GPS | max reported 180 | 0.1 | 0.15 |
| Wind direction (°) | estimated from GPS | 0 to 360 | 0.1 | 2 |

are the same as in Vaisala's RS-41 radiosondes (upsondes), which were also employed during EUREC[4]A from the BCO and four other ship-based platforms (Stephan et al., 2020). Table 1 provides a brief summary of the type, the resolution and the expected performance from the sensors used in the RD-41 and RS-41 sondes.

## 2.2   Sondes Deployment

A total of 1216 dropsondes were launched: 896 from the German High-Altitude Long Range aircraft (HALO) and 320 from
the National Oceanic and Atmospheric Administration's (NOAA) Lockheed WP-3D Orion N43-RF aircraft (P3). The P3 was operated as a part of the Atlantic Tradewind Ocean-Atmosphere Mesoscale Interaction Campaign (ATOMIC), which itself was a part of the EUREC[4]A campaign. Throughout this manuscript, we use the term EUREC[4]A to refer to both experiments.

Both aircraft used the Airborne Vertical Atmospheric Profiling System (AVAPS; UCAR/NCAR, 1993) with 8 simultaneous channels, for the operation of the dropsondes, as well as for the processing and quality control of collected data. For HALO,
the dropsondes are launched from a pneumatic chute controlled manually, which is located at the rear, starboard side of the aircraft, slightly oriented towards the bottom of the fuselage. For the P3, the drop point is near the center of the fuselage, with a little offset to the starboard side. HALO typically launched sondes at an altitude between 10–10.5 km, whereas the P3 did so typically at ∼7.5 km. Some P3 sondes were launched at ∼3 km, when the P3 was flying typical lawn-mower patterns (straight, parallel, long legs connected by shorter, perpendicular legs; parts of some visible in the north in Fig. 1) at low altitudes to
facilitate launching Airborne eXpendable BathyThermographs (AXBTs). The total number of dropsondes launched from the two aircraft per flight is given in Table 2.

Nearly ninety percent (∼87%) of the dropsondes launched reported data as expected, with partial data being recorded by a large percentage of remaining sondes. Only 51 (∼4%) sondes provided no usable data. Almost all of these 51 sondes failed because of an error in automatically detecting launch, the cause for which was later attributed to a manufacturing error in
certain batches of dropsondes (Vaisala, personal communication). Success rates for the other aspects of measurements are described in more detail in Section 3.

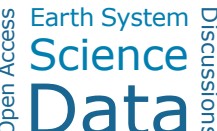

**Table 2.** Total number of dropsondes launched, circles flown during the flight, as well as takeoff and landing times (in UTC) for the flight are provided with corresponding flight IDs. Numbers in parentheses in the second column indicate the number of good dropsondes per flight (explained in § 3). Note that the table only shows circles with dropsonde launches. There were also circles flown with no dropsonde launches during EUREC$^4$A.

| Flight ID | Dropsondes (Good) | Takeoff time | Landing time | Circles with dropsondes |
|-----------|-------------------|--------------|--------------|-------------------------|
| P3-0117 | 23 (17) | 2020-01-17 14:00:02 | 2020-01-17 20:50:00 | 1 |
| P3-0119 | 28 (23) | 2020-01-19 13:25:25 | 2020-01-19 21:47:51 | 1 |
| P3-0123 | 38 (31) | 2020-01-23 13:21:02 | 2020-01-23 21:29:14 | 2 |
| P3-0124 | 16 (15) | 2020-01-24 13:21:05 | 2020-01-24 22:14:32 | 0 |
| P3-0131 | 25 (17) | 2020-01-31 15:32:27 | 2020-01-31 23:36:04 | 1 |
| P3-0203 | 22 (17) | 2020-02-03 13:19:02 | 2020-02-03 19:21:57 | 1 |
| P3-0204 | 31 (28) | 2020-02-04 13:19:52 | 2020-02-04 21:54:08 | 1 |
| P3-0205 | 29 (23) | 2020-02-05 13:22:23 | 2020-02-05 21:59:43 | 1 |
| P3-0209 | 32 (25) | 2020-02-09 01:56:28 | 2020-02-09 10:13:37 | 2 |
| P3-0210 | 32 (26) | 2020-02-10 01:48:57 | 2020-02-10 09:54:42 | 2 |
| P3-0211 | 44 (36) | 2020-02-11 03:15:07 | 2020-02-11 11:21:50 | 2 |
| HALO-0119 | 15 (12) | 2020-01-19 09:34:25 | 2020-01-19 18:48:03 | 1 |
| HALO-0122 | 73 (69) | 2020-01-22 14:57:35 | 2020-01-23 00:10:30 | 6 |
| HALO-0124 | 77 (71) | 2020-01-24 09:29:30 | 2020-01-24 18:41:13 | 6 |
| HALO-0126 | 75 (70) | 2020-01-26 12:05:30 | 2020-01-26 21:20:49 | 6 |
| HALO-0128 | 74 (71) | 2020-01-28 14:58:34 | 2020-01-28 23:55:17 | 6 |
| HALO-0130 | 4 (4) | 2020-01-30 11:19:34 | 2020-01-30 15:08:20 | 0 |
| HALO-0131 | 74 (68) | 2020-01-31 15:08:35 | 2020-01-31 23:56:53 | 6 |
| HALO-0202 | 89 (76) | 2020-02-02 11:28:02 | 2020-02-02 20:13:24 | 6 |
| HALO-0205 | 76 (65) | 2020-02-05 09:15:51 | 2020-02-05 18:21:22 | 6 |
| HALO-0207 | 73 (62) | 2020-02-07 12:02:24 | 2020-02-07 21:11:40 | 6 |
| HALO-0209 | 73 (66) | 2020-02-09 09:14:31 | 2020-02-09 18:03:00 | 6 |
| HALO-0211 | 61 (58) | 2020-02-11 12:29:05 | 2020-02-11 21:37:29 | 5 |
| HALO-0213 | 73 (69) | 2020-02-13 07:56:10 | 2020-02-13 17:17:17 | 6 |
| HALO-0215 | 51 (48) | 2020-02-15 15:07:30 | 2020-02-16 00:12:44 | 5 |
| HALO-0218 | 7 (1) | 2020-02-18 10:11:05 | 2020-02-18 18:55:31 | 0 |



A core part of the EUREC⁴A campaign was meso-scale circular flight patterns, which were adopted for most (1021 sondes, ∼84%) of the dropsonde launches. The use of a repetitive flight pattern was based on a desire to provide consistent and comparable estimates of meteorological variables. Circles were chosen to facilitate estimates of the profile of the meso (circle)

scale divergence of the horizontal wind. Following the error analysis of Bony and Stevens (2019) each circle aimed to launch twelve sondes. The number of sondes launched per circle is provided in Table 3.

Most circles were flown along a fixed circular path, called the EUREC⁴A-circle (Stevens et al., 2021), which was planned with the centre coordinates as 57.72°W, 13.30°N and a diameter of roughly 220 km. The location of the sonde launches shown in Fig. 1, highlight the density of HALO sondes concentrated along the circumference of the EUREC⁴A-circle. This circle

was chosen such that complementary measurements are maximised between the aircraft and other platforms in EUREC⁴A. Measurements performed along the EUREC⁴A-circle were made irrespective of meteorological conditions and hence were unbiased. Flight times (see Table 2) were adjusted to best sample the diel cycle given operational constraints. HALO was mostly restricted to daylight hours, while the P3 made 3 flights at night and is the only sampling of the night-time trades from EUREC⁴A dropsondes.

The actual mean diameter of all EUREC⁴A-circles marked by dropsonde launches was 222.82 km, and the mean centre was 57.67°W, 13.31°N. One circuit around the EUREC⁴A-circle took HALO roughly 60 min to execute at a flight level of about 9.5 km, resulting in sonde launches separated by about 5 min. There were 85 dropsonde circles flown during EUREC⁴A (see details in Table 3), and 73 of these were EUREC⁴A-circles, with HALO flying 70 of them and the rest flown by the P3. Of the 12 circles flown which were not EUREC⁴A-circles, one (HALO-0215_c3) was flown by HALO to provide spatial contrast for

comparison with measurements in the EUREC⁴A-circle. The remaining 11 non-EUREC⁴A circles were flown by the P3 and were mostly centered on the location of the NOAA research vessel Ronald H. Brown. The flight track of some of the P3 circles were approximated by a dodecagon.

Sondes were also dropped to sample conditions upwind and in the vicinity of EUREC⁴A-circles, to aid calibration of other instruments, as references for satellite underpasses, and to support surface based measurements from Research Vessels and

buoys. For instance, HALO typically separated a set of three standard EUREC⁴A-circles by an upwind 'excursion' toward the Northwest Tropical Atlantic Station buoy (NTAS) near 51.02°W, 14.82°N, along which 1 to 3 sondes were launched per flight.

Table 3: Details of circles flown during EUREC⁴A. Circle Time is the mean launch time for all sondes in the circle. Longitude (°E), latitude (°N) and diameter (km) are those associated with the center of a least-squares fitted circle to all sondes. Dropsondes show total number of sondes launched in each circle. The number in parentheses (L4) show the number of good sondes (explained in §3) used for regression in Level-4.

| Circle ID | Circle Time | Longitude | Latitude | Diameter | Dropsondes (L4) |
|---|---|---|---|---|---|
| P3-0117_ci1 | 15:55 | -51.00 | 14.84 | 181.86 | 12 (9) |
| P3-0119_ci1 | 15:02 | -52.97 | 14.50 | 180.90 | 12 (9) |
| | | | | | Continued on next page |

Table 3: Details of circles flown during EUREC$^4$A. Circle Time is the mean launch time for all sondes in the circle. Longitude (°E), latitude (°N) and diameter (km) are those associated with the center of a least-squares fitted circle to all sondes. Dropsondes show total number of sondes launched in each circle. The number in parentheses (L4) show the number of good sondes (explained in §3) used for regression in Level-4.

| Circle ID | Circle Time | Longitude | Latitude | Diameter | Dropsondes (L4) |
|---|---|---|---|---|---|
| P3-0123_ci1 | 14:31 | -54.96 | 14.38 | 186.96 | 12 (11) |
| P3-0123_ci2 | 20:12 | -55.68 | 13.29 | 185.34 | 12 (9) |
| P3-0131_ci1 | 16:53 | -54.38 | 13.84 | 184.74 | 12 (7) |
| P3-0203_ci1 | 14:40 | -54.50 | 13.92 | 183.19 | 13 (11) |
| P3-0204_ci1 | 14:50 | -53.14 | 13.49 | 184.45 | 12 (11) |
| P3-0205_ci1 | 15:12 | -53.26 | 12.23 | 179.63 | 12 (10) |
| P3-0209_ci1 | 04:55 | -57.67 | 13.26 | 243.92 | 12 (10) |
| P3-0209_ci2 | 06:22 | -54.87 | 13.84 | 187.64 | 12 (10) |
| P3-0210_ci1 | 04:55 | -57.74 | 13.30 | 220.41 | 12 (7) |
| P3-0210_ci2 | 06:12 | -54.78 | 13.77 | 184.77 | 12 (11) |
| P3-0211_ci1 | 06:06 | -57.71 | 13.30 | 221.59 | 12 (12) |
| P3-0211_ci2 | 07:16 | -55.49 | 14.23 | 185.69 | 13 (10) |
| HALO-0119_c1 | 17:53 | -57.86 | 13.27 | 186.65 | 12 (10) |
| HALO-0122_c1 | 15:45 | -57.70 | 13.27 | 222.45 | 12 (12) |
| HALO-0122_c2 | 16:58 | -57.71 | 13.28 | 223.80 | 12 (12) |
| HALO-0122_c3 | 18:09 | -57.72 | 13.27 | 220.95 | 12 (11) |
| HALO-0122_c4 | 20:12 | -57.70 | 13.29 | 224.13 | 13 (12) |
| HALO-0122_c5 | 21:27 | -57.71 | 13.29 | 223.64 | 12 (11) |
| HALO-0122_c6 | 22:34 | -57.71 | 13.31 | 225.99 | 12 (11) |
| HALO-0124_c1 | 10:19 | -57.62 | 13.23 | 231.01 | 13 (10) |
| HALO-0124_c2 | 11:30 | -57.66 | 13.28 | 224.16 | 13 (13) |
| HALO-0124_c3 | 12:42 | -57.69 | 13.28 | 223.83 | 12 (12) |
| HALO-0124_c4 | 13:57 | -57.68 | 13.27 | 223.80 | 12 (12) |
| HALO-0124_c5 | 15:07 | -57.69 | 13.27 | 223.98 | 12 (12) |
| HALO-0124_c6 | 16:16 | -57.68 | 13.27 | 223.47 | 13 (11) |
| HALO-0126_c1 | 12:49 | -57.68 | 13.29 | 224.27 | 12 (12) |
| HALO-0126_c2 | 14:01 | -57.67 | 13.30 | 219.53 | 12 (10) |



Table 3: Details of circles flown during EUREC⁴A. Circle Time is the mean launch time for all sondes in the circle. Longitude (°E), latitude (°N) and diameter (km) are those associated with the center of a least-squares fitted circle to all sondes. Drop-sondes show total number of sondes launched in each circle. The number in parentheses (L4) show the number of good sondes (explained in §3) used for regression in Level-4.

| Circle ID | Circle Time | Longitude | Latitude | Diameter | Dropsondes (L4) |
|---|---|---|---|---|---|
| HALO-0126_c3 | 15:10 | -57.69 | 13.28 | 221.75 | 12 (11) |
| HALO-0126_c4 | 17:47 | -57.67 | 13.29 | 224.49 | 12 (12) |
| HALO-0126_c5 | 19:00 | -57.69 | 13.28 | 221.71 | 12 (11) |
| HALO-0126_c6 | 20:19 | -57.67 | 13.29 | 224.16 | 12 (12) |
| HALO-0128_c1 | 15:46 | -57.70 | 13.30 | 223.86 | 12 (12) |
| HALO-0128_c2 | 16:57 | -57.70 | 13.30 | 223.84 | 12 (12) |
| HALO-0128_c3 | 18:11 | -57.69 | 13.32 | 221.09 | 12 (11) |
| HALO-0128_c4 | 20:25 | -57.71 | 13.32 | 226.18 | 12 (11) |
| HALO-0128_c5 | 21:41 | -57.70 | 13.30 | 223.89 | 12 (12) |
| HALO-0128_c6 | 22:55 | -57.70 | 13.30 | 224.00 | 13 (12) |
| HALO-0131_c1 | 15:57 | -57.70 | 13.31 | 225.87 | 11 (11) |
| HALO-0131_c2 | 17:06 | -57.72 | 13.31 | 226.37 | 12 (11) |
| HALO-0131_c3 | 18:20 | -57.69 | 13.30 | 219.07 | 12 (10) |
| HALO-0131_c4 | 20:28 | -57.69 | 13.29 | 224.03 | 12 (12) |
| HALO-0131_c5 | 21:42 | -57.70 | 13.29 | 224.15 | 12 (12) |
| HALO-0131_c6 | 22:54 | -57.68 | 13.29 | 221.11 | 12 (11) |
| HALO-0202_c1 | 12:12 | -57.72 | 13.29 | 224.80 | 12 (10) |
| HALO-0202_c2 | 13:18 | -57.71 | 13.28 | 223.70 | 12 (11) |
| HALO-0202_c3 | 14:27 | -57.71 | 13.27 | 225.77 | 13 (11) |
| HALO-0202_c4 | 16:55 | -57.70 | 13.28 | 226.26 | 12 (9) |
| HALO-0202_c5 | 18:03 | -57.72 | 13.28 | 222.28 | 12 (11) |
| HALO-0202_c6 | 19:06 | -57.71 | 13.29 | 224.82 | 13 (12) |
| HALO-0205_c1 | 09:59 | -57.70 | 13.29 | 220.90 | 12 (11) |
| HALO-0205_c2 | 11:11 | -57.70 | 13.28 | 220.83 | 14 (11) |
| HALO-0205_c3 | 12:21 | -57.72 | 13.26 | 223.22 | 12 (11) |
| HALO-0205_c4 | 15:03 | -57.71 | 13.23 | 224.79 | 12 (10) |
| HALO-0205_c5 | 16:11 | -57.73 | 13.28 | 226.25 | 13 (10) |





Table 3: Details of circles flown during EUREC[4]A. Circle Time is the mean launch time for all sondes in the circle. Longitude (°E), latitude (°N) and diameter (km) are those associated with the center of a least-squares fitted circle to all sondes. Dropsondes show total number of sondes launched in each circle. The number in parentheses (L4) show the number of good sondes (explained in §3) used for regression in Level-4.

| Circle ID | Circle Time | Longitude | Latitude | Diameter | Dropsondes (L4) |
|---|---|---|---|---|---|
| HALO-0205_c6 | 17:24 | -57.73 | 13.26 | 221.54 | 12 (11) |
| HALO-0207_c1 | 12:47 | -57.73 | 13.28 | 223.83 | 12 (12) |
| HALO-0207_c2 | 13:57 | -57.74 | 13.28 | 223.89 | 12 (11) |
| HALO-0207_c3 | 15:08 | -57.73 | 13.29 | 223.34 | 12 (7) |
| HALO-0207_c4 | 17:44 | -57.74 | 13.28 | 228.51 | 12 (10) |
| HALO-0207_c5 | 18:57 | -57.73 | 13.28 | 224.13 | 12 (12) |
| HALO-0207_c6 | 20:14 | -57.75 | 13.30 | 226.98 | 12 (9) |
| HALO-0209_c1 | 10:00 | -57.70 | 13.26 | 224.99 | 12 (11) |
| HALO-0209_c2 | 11:12 | -57.70 | 13.26 | 224.47 | 12 (10) |
| HALO-0209_c3 | 12:26 | -57.68 | 13.28 | 221.13 | 12 (11) |
| HALO-0209_c4 | 14:27 | -57.70 | 13.26 | 224.30 | 12 (11) |
| HALO-0209_c5 | 15:37 | -57.70 | 13.26 | 223.40 | 12 (11) |
| HALO-0209_c6 | 16:53 | -57.68 | 13.28 | 220.94 | 12 (11) |
| HALO-0211_c1 | 13:25 | -57.71 | 13.32 | 229.02 | 12 (10) |
| HALO-0211_c2 | 14:38 | -57.66 | 13.30 | 223.89 | 12 (12) |
| HALO-0211_c3 | 15:49 | -57.64 | 13.31 | 221.66 | 11 (11) |
| HALO-0211_c4 | 17:05 | -57.67 | 13.32 | 225.76 | 12 (11) |
| HALO-0211_c5 | 18:25 | -57.66 | 13.30 | 224.00 | 12 (12) |
| HALO-0213_c1 | 08:43 | -57.65 | 13.32 | 223.47 | 12 (12) |
| HALO-0213_c2 | 09:55 | -57.65 | 13.33 | 222.41 | 12 (10) |
| HALO-0213_c3 | 11:04 | -57.65 | 13.32 | 223.76 | 12 (12) |
| HALO-0213_c4 | 13:33 | -57.65 | 13.32 | 223.49 | 12 (12) |
| HALO-0213_c5 | 14:49 | -57.65 | 13.32 | 223.42 | 12 (12) |
| HALO-0213_c6 | 16:03 | -57.66 | 13.32 | 224.27 | 12 (11) |
| HALO-0215_c1 | 16:06 | -57.73 | 13.29 | 223.31 | 11 (11) |
| HALO-0215_c2 | 17:14 | -57.68 | 13.25 | 229.76 | 8 (7) |
| HALO-0215_c3 | 18:47 | -52.04 | 13.91 | 224.94 | 12 (11) |



Table 3: Details of circles flown during EUREC[4]A. Circle Time is the mean launch time for all sondes in the circle. Longitude (°E), latitude (°N) and diameter (km) are those associated with the center of a least-squares fitted circle to all sondes. Dropsondes show total number of sondes launched in each circle. The number in parentheses (L4) show the number of good sondes (explained in §3) used for regression in Level-4.

| Circle ID | Circle Time | Longitude | Latitude | Diameter | Dropsondes (L4) |
|---|---|---|---|---|---|
| HALO-0215_c5 | 22:10 | -57.67 | 13.33 | 222.44 | 13 (13) |
| HALO-0215_c6 | 23:12 | -57.59 | 13.22 | 237.56 | 7 (6) |

Additional details and strategies for HALO and P3 flights which may be informative for those sondes not launched on standard circles, can be found in Konow et al. (2021) and Pincus et al. (2021), respectively.

The maximum drift of the sondes from their launch locations in the horizontal space had a median of around $2.5\,\mathrm{km}$, as seen in Figs. 2 and 3. In the lower troposphere, the drift was generally more along the zonal direction than in the meridional direction, with sondes tending to drift towards the southeast of the launch location. Due to a climatological wind reversal at near 3 km the maximum displacement for HALO is at about this level, whereas for the P3 which dropped its sondes from a lower altitude and thus sampled less of the upper level westerlies, the maximum displacement is at the surface. This also explains why the drift of the P3 sondes is systematically to the west of the drop, and less directionally biased for the HALO sondes. The P3 sondes typically sampled the sub-cloud layer $\sim 0.03°$ southwest of the launch location, whereas for HALO sondes, the direction of drift was influenced strongly by the winds above $3\,\mathrm{km}$, and therefore varied between different flight days.

## 2.3 Raw Data and Initial Processing

The raw data collected on the aircraft by AVAPS and the subsequent processing with the Atmospheric Sounding Processing Environment (ASPEN; Martin and Suhr, 2021) software constitute Levels 0 and 1 of *JOANNE*, respectively. The data included as part of these two levels involve no external adjustments other than the standard processing and quality control by AVAPS and ASPEN – both state-of-the-art tools for dropsonde measurements.

### 2.3.1 Level-0 (Raw data)

Level-0 includes the raw files generated by AVAPS during dropsonde measurements. For every dropsonde launch, multiple files are generated, which store the collected data in different formats, with there being some extent of information overlap between them. These files have names starting with a capitalised letter and are described in Table 4 with the corresponding letter as the file type.





**Table 4.** Table shows file types included in Level-0, which are all files in the raw data collected by the dropsondes, and a brief description of what they entail.

| File Type | Description |
| --- | --- |
| A Files | Sounding attributes file; includes channel configuration, COM ports data, hardware configuration, launch obs data, sensor errors, aircraft data, software config and firmware information |
| B Files | File containing binary data; same as D-Files |
| C Files | Sounding data stored as comma-separated-value files |
| D Files | Raw sounding data recorded for timestamp at every $0.25\,\mathrm{s}$ |
| D_P Files | Only post-launch raw data; same as D-Files |
| R Files | Receiver ports data: signal strength and receiver frequency |
| 0_SysLog Files | comma-separated-value file of all AVAPS system logs |
| 1_Aircraft Files | TXT file of aircraft position data in the IWGADTS Format (IWG1) |
| 2_GPSRef Files | TXT file of GPS data: GPGGA (system fix data) and GPRMC (minimum specific GPS/Transit data) |
| 3_SpecAnlyzr Files | TXT file of logs of spectral analyzer |

In addition to these files, information about the hardware and the aircraft data are generated and stored each time the AVAPS system is switched on, usually once per flight. These files have names preceded by a number, and the type and content of these
120 files are given in Table 4.

All Level-0 files of a single day (as per UTC) are stored in their respective date directories, with their names in the format `YYYYMMDD`. The P3 and HALO directories are separated into two different directories named after the respective aircraft.

### 2.3.2 Level-1 (ASPEN processed data)

Level-1 includes all files from Level-0 after processing by ASPEN. ASPEN takes in D-type files (see Table 4) as input, and gives
an output of quality controlled files. For *JOANNE*, the D files were supplied as input to BatchASPEN v3.4.3, and all output files have the suffix `_QC`. The files are in NetCDF format. For ASPEN processing, we used the standard *editsonde* configuration. A detailed explanation of the file-structure of these `_QC.nc` files and the processing steps carried out by ASPEN are outlined in detail by Martin and Suhr (2021). These Level-1 `_QC.nc` files serve as the input for further processing in *JOANNE*.

## 3 Quality Control (QC)

For the data products post Level-1, *JOANNE* aims to provide sounding profiles that do not contain any obvious measurement errors and contain minimal missing data records. After the ASPEN processing, we run additional QC tests on all Level-1 sounding profiles and filter out soundings that do not meet these objectives. Profiles which are filtered out during this QC are

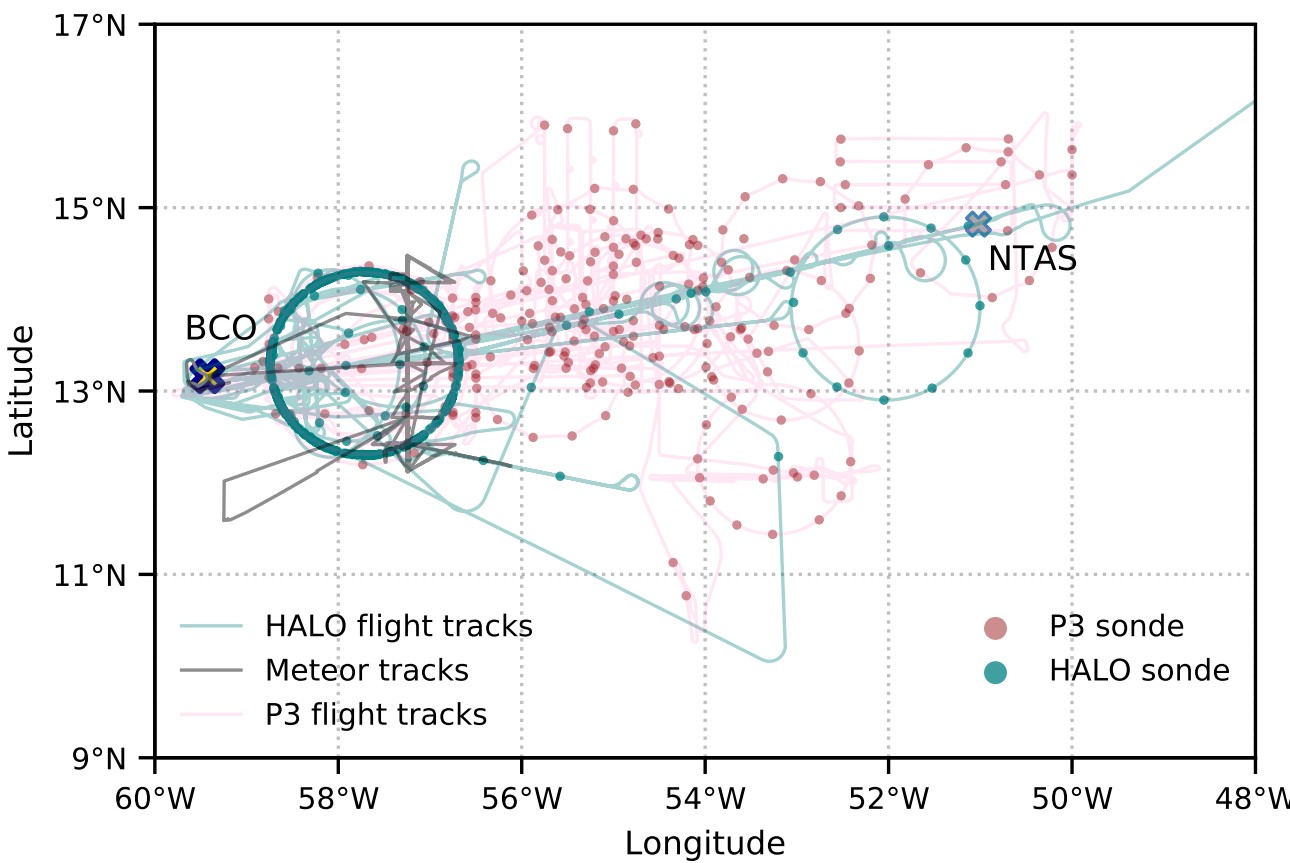

**Figure 1.** Map showing the launch locations of the dropsondes during EUREC$^4$A from HALO (teal) and P3 (red). The flight paths for HALO (light teal), P3 (light pink) and Meteor (gray) are shown as shaded lines. The crosses near the west and east edges of the displayed domain mark the location of the BCO and the NTAS buoy, respectively.

not included in Level-2 and onwards. We believe that soundings passing such a QC stage would best fulfil the purpose of the dropsondes – to characterise the EUREC$^4$A atmospheric environment – with little to no troubleshooting at the user end.

However, users who wish to pursue a specific measurement that did not make it past the QC stage can still avail themselves of it in the exhaustive Level-0 and Level-1 data products.

A sounding's success in the QC stage is provided by a parameter *qc_flag* which has possible values of *good*, *bad* and *ugly*. The values stand for fully-usable, non-usable and partially-usable data, respectively and are described in more detail later with relevant context. Only soundings flagged as *good* are included in *JOANNE* after Level-1. A sounding's *qc_flag* value is

140 determined by its collective performance in three tests that are designed with the aforementioned QC objectives in mind. These tests are listed as follows.

   1. Launch Detection Test (*ld_test*) : This test filters sondes that failed to detect an automatic launch

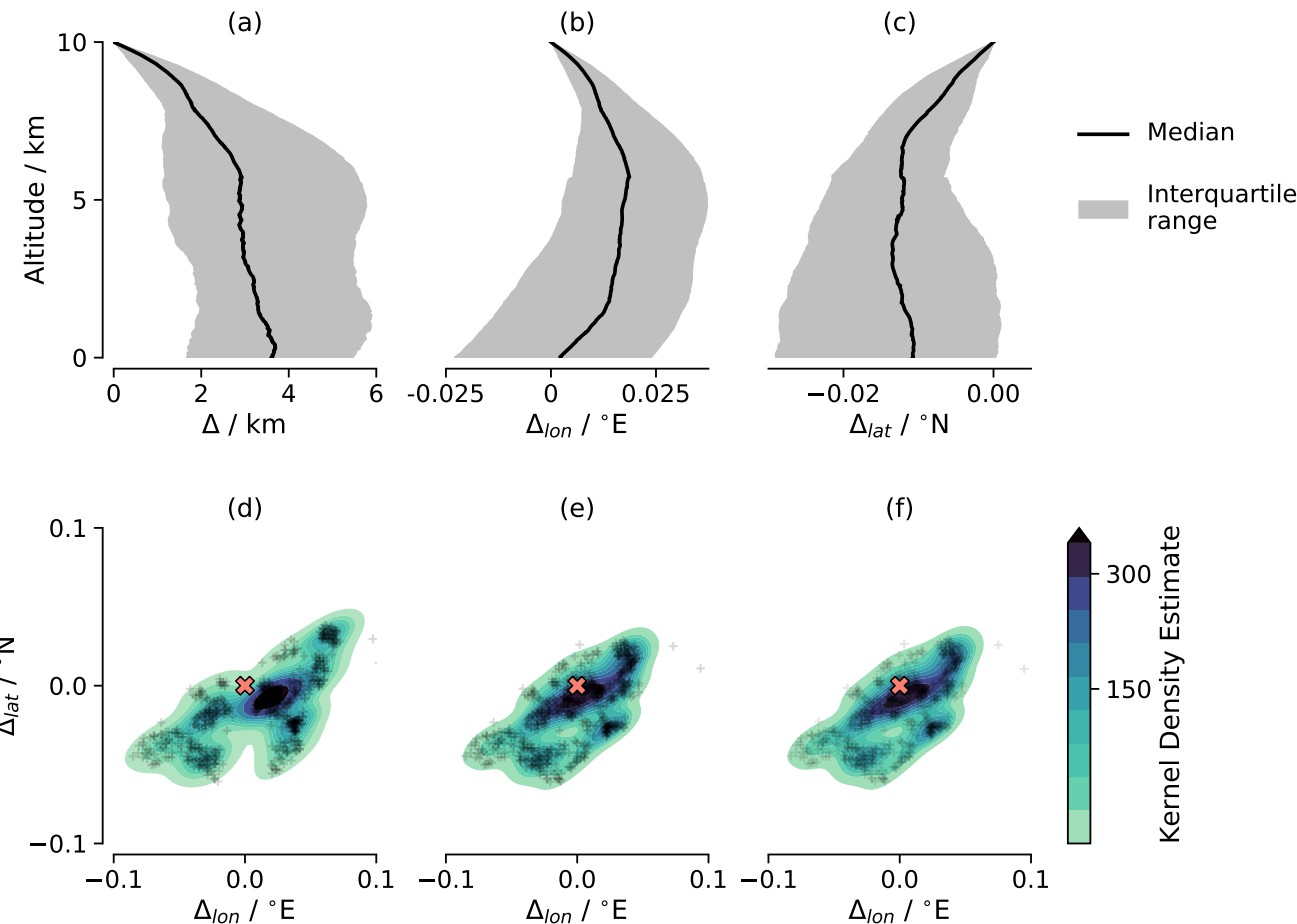

**Figure 2.** Figure shows an overview of the drift in HALO sondes. $\Delta$ indicates horizontal displacement in sondes from launch location. Figures (a)-(c) show the median drift from launch and the corresponding interquartile range for (a) horizontal displacement, (b) longitude and (c) latitude. Figures (d)-(f) show as colours the kernel density estimates (KDE) of drift from launch location (red cross) at (d) median altitude of maximum drift in the profile, $\tilde{z}=3140$ m (e) sub-cloud layer mean (0-500 m) and (f) at altitude of 2 km, where usually the cloud-top layer is present.

**Figure 3.** Same as Fig. 2, but for P3 sondes instead of HALO sondes. For (d), median altitude of maximum drift in the profile, $\tilde{z}=10\,\text{m}$

2. Profile Fullness Test (*sat_test*) : This test filters sondes that did not record measurements for at least 80% of the time measured in the profile

3. Low-altitude Measurements Test (*low_test*) : This test filters sondes whose measurements in the lower levels of the atmosphere do not fall within the expected bounds of parameter values.

The details of how a sounding's performance is judged with these tests and how these tests combine to give the *qc_flag* value for the sounding are explained further in this section.

### 3.1   Launch Detection Test (*ld_test*)

This test checks whether the sonde detected a launch automatically. If a sonde fails to automatically detect a launch, it does not switch to high-power signal transmission, and thus, fails to send data back to the AVAPS PC in the aircraft, after it has passed





further than a short range. The receiver in the aircraft failed to detect any signal from such sondes usually after they had fallen below pressure levels of $300\,\mathrm{hPa}$.

A sounding's success in this test is marked by a parameter of the same name, i.e. *ld_test* and has possible values of 0 and 1, which correspond to *bad* and *good*, respectively. The primary method to check launch detection is to parse through the sounding attribute log files (A-type; see Table 4) in Level-0. These files have names starting with 'A' and are followed by the date and time of launch. The file extension is the number of the channel used to initialise the sonde and receive its signal. Note that for sondes that did not detect a launch, the file name has time when the sonde was initialised, whereas for the rest, the file name is for the time of the detected launch. The log file contains an internal record termed 'Launch Obs Done?'. If this value

is 1, the launch was detected, else if it is 0, launch was not detected. The same values are used to mark the *ld_test*.

## 3.2   Profile Fullness Test (*sat_test*)

This test checks the abundance of measurements within a sounding profile relative to the flight time of the sonde. For a raw measurement profile, time is the independent dimension along which records of measurements are made. The time record is given by the $4\,\mathrm{Hz}$ GPS measurements which means that for the $2\,\mathrm{Hz}$ PTU measurements every other record is a missing value.

Ideally, all parameters (except u,v) will have measurements at every other time record, and u,v at every time record, but in practice, the number of records with measurements always falls short of the ideal number. This is because the time records also include values during initialisation as well as during a little before and after the launch, when no signal can be sent back to the AVAPS PC. Thus, the ratio of actual measurements to total possible measurements is lower than the ideal estimate of 1.

    The profile fullness test is run by checking the abundance of measurements individually for all parameters in a sounding. The

success of the test for a parameter $\phi$ is recorded in a corresponding parameter $\phi\_test$, e.g. *p_test* corresponding to *p* (pressure) and this success is determined by the ratio of the count of its measurements ($n$) to its total possible measurements ($N$), denoted by,

$$\phi_{\mathrm{sat}} = \frac{n(\phi)}{N(\phi)}. \tag{1}$$

Accounting for the different sampling rates of the GPS and PTU measurements, the distributions of $\phi_{\mathrm{sat}}$ is shown in Fig. 4, which shows that peaks start to flatten below 0.8. Thus, we set a threshold value of 0.8, and if parameter $\phi$ has $\phi_{\mathrm{sat}}$ lower than this threshold, then it is taken as not having a complete profile, and $\phi\_test$ is flagged as *ugly*. If $\phi_{\mathrm{sat}}$ exceeds or matches the threshold, $\phi\_test$ is flagged as *good*. If all values are missing, i.e. $\phi_{\mathrm{sat}} = 0$, then $\phi\_test$ is flagged as *bad*.

    Whereas the aforementioned tests ($\phi\_test$) recorded the success for every parameter in a sounding, we use *sat_test* to record

the success of a sounding. For a given sounding, if all parameter tests are *good*, the sounding's *sat_test* is flagged as *good*. Similarly, if all individual parameter tests are *bad*, *sat_test* is flagged as *bad*. If neither of these conditions is met, *sat_test* is flagged as *ugly*.




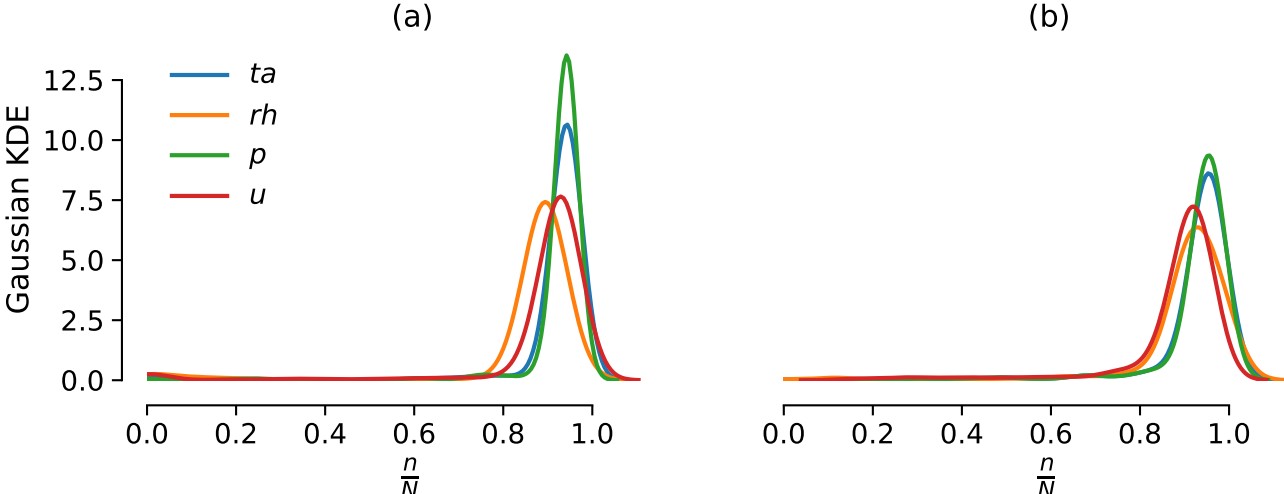

**Figure 4.** Kernel density estimate of ratio of actual measurement counts ($n$) out of maximum possible count of measurements ($N$), based on the timestamp records in each sonde Level-1 file for (a) HALO and (b) P3. For $u$, $N$ would be the total timestamp records in any given sonde profile, whereas for the rest it would be half that. In the legend, labels stand for temperature ($ta$), relative humidity ($rh$), pressure ($p$) and eastward wind ($u$). The northward wind ($v$) has the same distribution as $u$, and is hence not shown.

### 3.3 Low-altitude Measurements Test (*low_test*)

This test functions as a sanity-check for the measurements from a sounding in the lower levels of the atmosphere, which is mostly near the surface except for one test where the check is for the lowest $4\,\mathrm{km}$. Similar to the profile fullness test, this test is also determined by the success of parameters over different individual tests. The success of these individual tests are recorded with a parameter name same as that of the corresponding test name. For each of these tests, if the sounding passes the test, it is marked as *good*, else as *bad*. The individual tests and their criteria for passing are as follows.

1. *low_p_test*

    This test checks if maximum pressure measured in a sounding is within bounds ($1000$–$1020\,\mathrm{hPa}$) and if so, the sounding passes the test. If maximum value of $p$ is greater than upper bound, it is unrealistic, and if lesser than lower bound, it means that the sonde did not measure the near-surface levels of the atmosphere. This test does not check any GPS values. Even if there were no pressure measurements higher than $1000\,\mathrm{hPa}$, there may still be GPS measurements in the low-altitude levels. Such sondes can still be useful for wind and wind-derived products.

2. *low_t_test*

    This test checks if air temperature measured in a sounding is within bounds. It sets two criteria for bounds, which are (a) maximum air temperature recorded should not be greater than $30\,^\circ\mathrm{C}$ and (b) mean $T$ in the bottom $100$ m should not be lesser than $20\,^\circ\mathrm{C}$. If either of the above limits is violated, measurement of $T$ for the sounding is considered out of





bounds, and marked as *bad*. The sonde is also marked *bad*, if there are no measurements in the bottom 100 m (by GPS altitude (*gpsalt*) in Level-1).

3. *low_rh_test*

   This test checks if relative humidity measured in a sounding is within bounds. The criterion is that mean RH in the bottom $100\,\mathrm{m}$ should not be lesser than 50%. If this bound is violated, RH for the sonde is considered out of bounds, and marked as *bad*. The sonde is also marked *bad*, if there are no measurements in the bottom $100\,\mathrm{m}$.

4. *low_z_test*

   This test checks if minimum *gpsalt* of a sounding is within bounds, i.e. $\leq 30\,\mathrm{m}$ above mean sea level. A value higher than the bound means there are no near-surface measurement values of GPS and consequently, horizontal winds. This flag does not include any geopotential height values. Even if there are no GPS values below $30\,\mathrm{m}$, there may still be PTU measurements in the lowest levels.

5. *palt_gpsalt_rms_test*

   This test checks if the root mean square (RMS) difference between geopotential altitude (palt) and the GPS altitude (gpsalt),for values below $4\,\mathrm{km}$, and is lower than $100\,\mathrm{m}$. If the estimated RMS difference is below the limit, then the sounding is flagged as *good*. If the estimated RMS difference is greater than the limit, or if there are no values of either *palt* or *gpsalt* overlapping in the lower 4 km, then the sounding is flagged as *bad*. The lack of overlap could be because either there are no palt values or no gpsalt values or both.

Based on the success in the aforementioned individual tests, the overall success of a sounding for the low-altitude measurements test is recorded in the parameter *low_test*. If all individual tests are flagged as *good*, the *low_test* is flagged as *good*, and similarly, if all individual tests are flagged as *bad*, the *low_test* is flagged as *bad*. If neither of these conditions is met, the sounding's *low_test* is flagged as *ugly*.

Note that the bounds used for the individual tests are all considered keeping in mind the EUREC[4]A region and conditions. For a similar QC in a different region or environment, the bounds for the parameters will likely be different.

## 3.4  *qc_flag*

The overall success of a sounding is recorded as values of *good*, *bad* or *ugly* in the *qc_flag* parameter, and is determined by the combination of success through the three QC tests, as shown in Table 5.

**Table 5.** Determination of *qc_flag* value based on success of sounding in the three QC tests – *ld_test*, *sat_test* and *low_test*. The * indicates that any value for the test satisfies the condition.

| ld_test | sat_test | low_test | qc_flag |
|---------|----------|----------|---------|
| *good* | *good* | *good* | **good** |
| *bad* | * | * | **bad** |
| * | *bad* | *bad* | **bad** |
| – All other combinations – | | | **ugly** |

Table 6 summarises the statistics of the QC tests for HALO and P3. Although the process of classifying the sondes can be simplified by other combinations of the *sat_test* and *low_test* values, the method we present ensures no good sondes are omitted, and no bad sondes are admitted. The rest of the sondes, the ugly sondes, still have data that can be salvaged, and after some additional QC, can be combined with the other good sondes depending on the user's objective.

*JOANNE* provides a status file per platform, which stores the results for each individual test and group of tests mentioned
above, as well as the final *qc_flag* classification for each sounding. Thus, the user can still mould the classification based on their objectives, add or remove tests to the process and customise the sonde selection for themselves.

**Table 6.** Count of sondes that passed each QC test, separated by platforms

| Platform | Classification | ld_test | sat_test | low_test | qc_flag |
|----------|---------------|---------|----------|----------|---------|
| HALO | Good | 854 | 814 | 831 | **810** |
| | Bad | 41 | 1 | 40 | **41** |
| | Ugly | N/A | 80 | 24 | **44** |
| P3 | Good | 312 | 270 | 280 | **258** |
| | Bad | 4 | 0 | 8 | **10** |
| | Ugly | N/A | 52 | 34 | **54** |

## 4   Dry bias in HALO dropsondes

The radiosonde measurements during EUREC[4]A taken from the BCO and the research vessel Meteor show evidence of a dry bias in the humidity measurements of the HALO dropsondes. The HALO measurements are bounded by Meteor's on the
upwind side and BCO's on the downwind side (see Fig. 1). Since all three platforms have unbiased sampling, we expect that the HALO distribution should be between the other two. Fig. 5 shows that the BCO and Meteor distributions of RH align closely throughout the lower troposphere and thus, HALO measurements should not differ. The offset in the HALO measurements towards lower RH values suggests a dry bias in the HALO sondes. Since the sensors for the dropsondes and the radiosondes are the same, an instrument difference can be ruled out. Further comparisons with other water vapour measurements in the



vicinity such as the radiosondes from the ship Ron Brown, surface humidity measurements from both ships and dropsondes from the P3 aircraft also show HALO's median specific humidity to be lower than expected (not shown).

    A possible contamination of the polymer film in the moisture sensor could affect its dielectric constant, whose fluctuations with respect to relative humidity is subsequently affected. The most plausible explanation is that the reconditioning procedure of HALO dropsondes was improper, which resulted in some trace gas pollutants being retained on the humidity sensor, and

should otherwise have been removed during the reconditioning. The protocols of operation for P3 and HALO were not the same, and this leads us to believe that the improper reconditioning was only an issue for the dropsondes launched from HALO. In the case of radiosondes, the reconditioning is part of the automatic calibration process, and so it is not expected to cause problems.

    A multiplicative correction factor of 1.06 to the RH values (dotted line in Fig. 5) aligns the HALO distributions well with

the BCO and Meteor distributions. The success of this simple rescaling, in both matching the mean and the variance of the distributions, suggests that the bias is both multiplicative and systematic. Had the bias come from a subset of the sondes, a multiplicative correction to match the mean would have resulted in a broader distribution. Had the bias been an additive one, then the correction would have not been as successful at all heights. It is not, however, understood how the contamination of the sensor leads to this dry bias, and why the multiplicative correction appears to work so well. For these reasons, the correction is

not applied to the published data.

    Users of the data should also be aware that the uncorrected dry bias in HALO dropsonde measurements will propagate into other variables, especially PW and moisture gradients (since it is apparently multiplicative) and will even have a slight effect on estimates of geopotential altitude which depend on the atmospheric density, and hence moisture content. However, the proposed multiplicative correction, should users wish to adopt it, is straightforward to apply to these data.



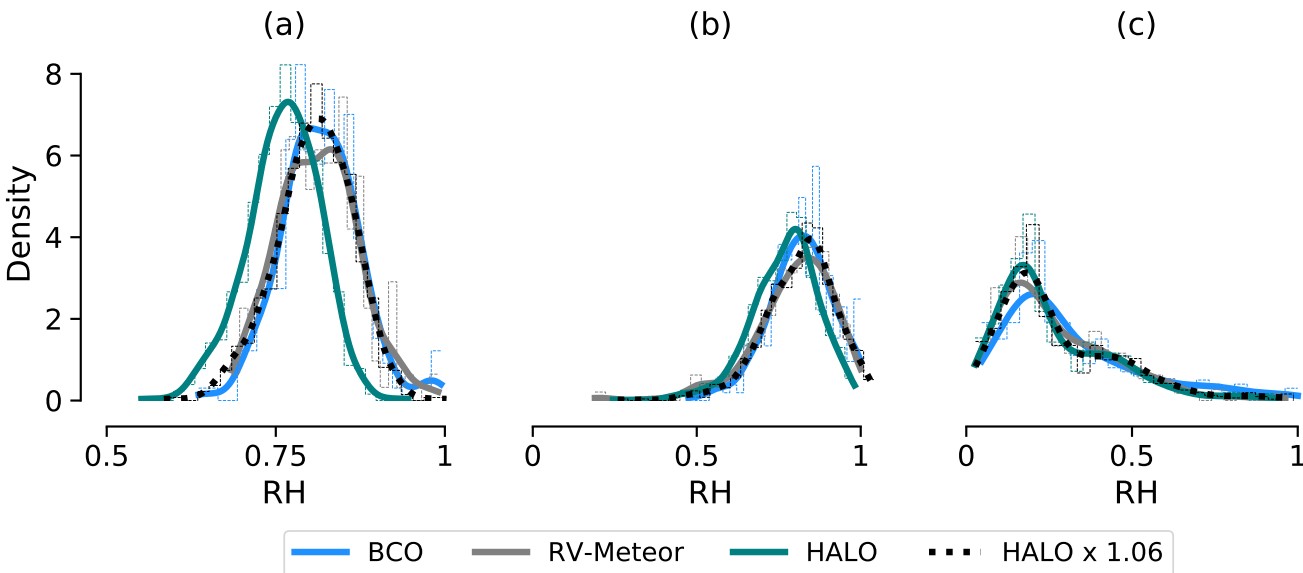

**Figure 5.** Spread (kernel density estimates) in relative humidity values from soundings made by BCO, Meteor and HALO at (a) mean of 0-500 m, (b) mean of 750-1500 m and (c) mean of 2000 - 4000 m. The last item in the legend is for RH values of HALO multiplied by 1.06. To coincide with HALO measurement times, BCO and Meteor soundings between 03:00 and 09:00 UTC have been excluded from these distributions, which has a relatively insignificant impact.

## 5 Data Products

### 5.1 Level-2 (Quality Controlled Sounding Data)

The Level-2 NetCDF files contain data from individual soundings, which passed with a *qc_flag* value of *good* from the QC stage (discussed in Section 3). For Level-2, only variables that are measurements from the dropsonde sensors are included. Redundant state variables are not carried forward from the Level-1 files. Products up to Level 2 maintain the raw measurement profile, and data variables are aligned along the independent dimension `time`.

File names in Level-1 are generally indicative of launch times, however for sondes that did not detect a launch, the file name indicates time of initialisation. The attribute `Launch-time-(UTC)` in every sounding file of Level-2 should be considered as the final authority on launch time. This is the same as the variable *launch_time* in Levels-1 and 3.

`sonde_id` is a variable available in *JOANNE* products from Level-2 onwards. This is a unique, immutable identifier and is meant to identify exactly one dropsonde which corresponds to exactly one sounding profile. Note that the identifier variable `sounding_id` in the EUREC[4]A radiosondes dataset (Stephan et al., 2020) identifies sounding trajectory and not instrument, since one instrument can have upward and downward trajectories. The *JOANNE* variable `sonde_id` functions solely as an identifier and no information should be interpreted from the semantics of this variable.





**Table 7.** Table shows the structure for the Level-2 product, outlining the coordinates, variables and their corresponding descriptions, units and dimensions.

| OBJECT | NAME | DESCRIPTION | UNITS | DIMENSION |
|---|---|---|---|---|
| Coordinates | time | time of recorded measurement | seconds since 2020-01-01 | time |
| | alt | geopotential height | m | time |
| | lat | latitude | degree_north | time |
| | lon | longitude | degree_east | time |
| Variables | p | atmospheric pressure | Pa | time |
| | ta | air temperature | K | time |
| | rh | relative humidity | | time |
| | wspd | wind speed | m s$^{-1}$ | time |
| | wdir | wind direction | degrees | time |
| | sonde_id | sonde identifier | | |

The Level-2 product consists of individual files for every sounding with the file structure as shown in Table 7. All files also include flight information such as position, height, and speed as attributes. These are saved by the AVAPS aircraft computer in the sonde A-files (see Table 4) and is input from the aircraft system itself. The files also have additional attributes such as the software version for post-processing and quality control. The file names are in the format:

`[campaign]_[project]_[instrument]_[sonde_id]_[version].nc`,

e.g. `EUREC4A_JOANNE_Dropsonde-RD41_HALO-0124_s42_v0.11.0.nc`. Note that `sonde_id` includes one underscore character within its value for the example shown.

## 5.2 Level-3 (Gridded Data)

Level 3 is a product combining dropsonde measurements launched from both the HALO and P3 aircraft, interpolated onto a uniform vertical grid of 10 m spacing, similar to the processing of EUREC$^4$A radiosounding profiles (Stephan et al., 2020). The product is a single file which contains, all dropsondes from Level-2 along the altitude dimension `alt`.

### 5.2.1 Gridding

The primary objective behind the Level-3 product is gridding all soundings on a common, vertical grid, thus making it easier to use the soundings for different analyses. The vertical grid spacing for the dataset is kept at 10 m, up to an altitude of 10 km.

In the case of a regular drop, i.e. if there are no issues like a fast fall, or a failed parachute, the average descent rate of the dropsondes is $\sim$21 m s$^{-1}$ at 12 km altitude and $\sim$11 m s$^{-1}$ near to the surface. The PTU sensors have a measurement frequency of 2 Hz, while the GPS has a 4 Hz measurement frequency. This would translate to a vertical sampling of roughly 9–10 m at HALO's flight altitude, and 5–6 m close to the surface for the PTU values and correspondingly finer vertical sampling for the





GPS-based measurements. Hence the data are slightly coarsened, and only for PTU values in the upper-mid troposphere do the interpolated values exceed the resolution of the measurements. The gridding is carried out through the following steps.

1. Variables `q` (specific humidity), `theta` (potential temperature), `u` (eastward wind), and `v` (northward wind) are computed and added to the dataset (for details, see Section 5.2.2).

2. All variables along the height coordinate in the dataset are averaged on $10\,\mathrm{m}$ bins up to $10\,\mathrm{km}$ altitude. In cases where no data are available in the altitude bin, a linear interpolation from neighbouring measurements along the height dimension is used to estimate the value in the altitude bin, with the restraint that the neighbouring measurements are not further apart than $50\,\mathrm{m}$. If data are not available within $50\,\mathrm{m}$ of the desired height level, values at that height level are assigned `_FillValue`. While this still allows for a few missing values ($\sim$2-3 considering a fall speed of $15$–$20\,\mathrm{m\,s^{-1}}$, it does not lead to substantial artificial information created by the smoothened interpolation between points relatively farther away.

3. Pressure values are interpolated logarithmically and these values replace the linearly interpolated pressure values.

4. Temperature ($T$) and relative humidity ($RH$) are the originally measured properties by the dropsonde sensors. However, for interpolation $q$ and $\theta$ are preferred, as these variables are conserved. After interpolation, $T$ and $RH$ are recomputed from the interpolated values of $\theta$ and $q$. The recomputed values for $T$ and $RH$ replace the previously interpolated $T$ and $RH$ variables from the sounding.

5. Wind speed and wind direction are computed from the interpolated values of $u$ and $v$ and added to the interpolated dataset.

Table 8: Table shows the structure for the Level-3 product, outlining the coordinates, variables and their corresponding descriptions, units and dimensions.

| OBJECT | NAME | DESCRIPTION | UNITS | DIMENSION |
|---|---|---|---|---|
| Coordinates | alt | height obtained by integrating upwards the atmospheric thickness estimated from the hypsometric equation | m | alt |
| | sonde_id | unique sonde ID | | sonde_id |
| | launch_time | time of dropsonde launch | seconds since 2020-01-01 | sonde_id |
| | interpolated_time | value of time (original independent dimension) linearly interpolated to altitude grid | seconds since 2020-01-01 | sonde_id, alt |





Table 8: Table shows the structure for the Level-3 product, outlining the coordinates, variables and their corresponding descriptions, units and dimensions.

| OBJECT | NAME | DESCRIPTION | UNITS | DIMENSION |
|---|---|---|---|---|
| | lat | latitude | degree_north | sonde_id, alt |
| | lon | longitude | degree_east | sonde_id, alt |
| Variables | p | atmospheric pressure | Pa | sonde_id, alt |
| | ta | dry bulb temperature | K | sonde_id, alt |
| | rh | relative humidity | | sonde_id, alt |
| | wspd | wind speed | $m\ s^{-1}$ | sonde_id, alt |
| | wdir | wind direction | degree | sonde_id, alt |
| | u | u-component of the wind | $m\ s^{-1}$ | sonde_id, alt |
| | v | v-component of the wind | $m\ s^{-1}$ | sonde_id, alt |
| | theta | air potential temperature | K | sonde_id, alt |
| | q | specific humidity | $kg\ kg^{-1}$ | sonde_id, alt |
| | low_height_flag | flag if flight height < 4 km when dropsonde was launched | | sonde_id |
| | platform_id | platform from which dropsonde was launched | | sonde_id |
| | flight_altitude | altitude of the aircraft when dropsonde was launched | m | sonde_id |
| | flight_lat | north latitude of the aircraft when dropsonde was launched | degree_north | sonde_id |
| | flight_lon | east longitude of the aircraft when dropsonde was launched | degree_east | sonde_id |
| | N_p | number of observations used to derive level 3 pressure data | | sonde_id, alt |
| | N_ta | number of observations used to derive level 3 temperature data | | sonde_id, alt |
| | N_rh | number of observations used to derive level 3 relative humidity data | | sonde_id, alt |
| | N_gps | number of observations used to derive level 3 GPS-data | | sonde_id, alt |





Table 8: Table shows the structure for the Level-3 product, outlining the coordinates, variables and their corresponding descriptions, units and dimensions.

| OBJECT | NAME | DESCRIPTION | UNITS | DIMENSION |
|---|---|---|---|---|
| | m_p | method used to derive Level-3 pressure data | | sonde_id, alt |
| | m_ta | method used to derive Level-3 temperature data | | sonde_id, alt |
| | m_rh | method used to derive Level-3 relative humidity data | | sonde_id, alt |
| | m_gps | method used to derive Level-3 GPS-data | | sonde_id, alt |
| | alt_bnds | cell interval bounds for altitude | m | alt, nv |

### 5.2.2 Added Variables

The complete list of variables, their units and dimensions for Level-3 are provided in Table 8. The descriptions of variables added in Level-3 are as follows.

Launch Time (`launch_time`)

Level-3 data are of the trajectory type with a single timestamp associated with each sounding, i.e. the launch time. This variable is the same as `launch_time` present in all Level-1 files.

Potential Temperature, $\theta$ (`theta`) and Specific Humidity, $q$ (`q`)

For estimating $\theta$, we consider standard pressure, i.e. $1000\,\mathrm{hPa}$. For the estimation of saturated vapour pressure, the method by Hardy (1998) is used with temperature at every altitude level as input and subsequently, specific humidity ($q$) is estimated. The values of $\theta$ and $q$ are estimated from the soundings on their respective, raw vertical grid, before interpolating them on to a common grid.

Platform Name (`platform`)

Although all soundings are in a single file in Level-3, they can still be separated into HALO and P3 sondes, using this variable, which specifies the platform from which the dropsonde was launched. The values of the variable are strings, and have two possible values – "HALO" and "P3".

Interpolated time (`interpolated_time`)

Since time is the independent dimension along which the measurements are made, it is illogical to average or interpolate time along the altitude dimension. Therefore, `time` is not available as a variable from Level-3 onwards. However,





for practical purposes, this can be useful information for instance, to compare with remote-sensing instruments on the aircraft. Thus, relying on the high sampling rate and based on the robust assumption that the dropsondes have negligible upward motion, Level-3 includes the variable `interpolated_time`. The variable is computed with linear interpolation, same as for other variables except pressure.

**Low Flight Height Flag (`low_height_flag`)**

Some of the sondes from the P3 were launched at an altitude of $\sim3\,\mathrm{km}$ when the aircraft was also launching AXBTs. Therefore, these soundings sampled only the lower levels of the atmosphere, over just half of the depth sampled by other P3 sondes, and a third of that of HALO's typical sondes. The `low_height_flag` variable in Level-3 marks sondes that have a launch altitude of less than $4\,\mathrm{km}$, with a value of 1 and otherwise, 0. This flag is useful to put in to context estimates of integrated quantities such as total column moisture, as well as to act as an easy separator for users who want to look at profiles in the free troposphere.

**Number of Measurements in Bin (`N_p`, `N_ta`, `N_rh`, `N_gps` and Bin Method (`m_p`, `m_ta`, `m_rh`, `m_gps`))**

The variables `N_p`, `N_ta`, `N_rh` and `N_gps` provide the number of pressure, temperature, relative humidity and GPS measurements, respectively in each altitude bin for gridding. Depending on the values of these N-variables, the corresponding cell methods – denoted by the m-variables – are provided. For the m-variables, possible values are 0, 1 and 2 and stand for no data, interpolation and averaging, respectively.

## 5.3 Level-4 (Circle Products)

As discussed in Section 2.2, the estimation of area-averaged mesoscale properties, such as divergence, was the primary objective behind the sondes' deployment over circular patterns. The Level-4 product provides these circle products as gradient terms estimated by regressing the parameters at each level for a set of sondes comprising a circle. Level-4 also includes terms of divergence, vorticity, vertical velocity and pressure velocity, which are subsequently computed from the gradient terms. The input data are from the gridded dataset in Level-3.

### 5.3.1 Identifying circles and corresponding sondes

The flight phase segmentation (FPS) files for HALO (Konow et al., 2021) and for P3 (Pincus et al., 2021) are used for identifying the circles and the dropsondes corresponding to these circle segments. To facilitate ease of working with *JOANNE* and the FPS files, the circle segments in *JOANNE* Level-4 have been tagged with the same segment IDs as those in the FPS files. Moreover, the FPS files include a list of dropsondes associated with every flight segment, and this list is comprised of sonde IDs that are the same as that in the *JOANNE* Level-3 gridded product.





### 5.3.2 Regression

Following Bony and Stevens (2019), for any parameter $\phi$ measured by a dropsonde, assuming that variation at any altitude level is linear in horizontal space and is steady in time, the value at any point can be estimated as,

$$\phi(x,y) \approx \phi_o + \frac{\partial \phi}{\partial x}\Delta x + \frac{\partial \phi}{\partial y}\Delta y, \tag{2}$$

where $\phi_o$ is the meso-scale mean value, $\Delta x$ and $\Delta y$ are the eastward and northward distances respectively, from the mean center-point of all observed points included in the regression. Minimising the least-squared errors for the linear regression fit shown in Eq-2, would give an estimate of the linear variation in the eastward ($\frac{\partial \phi}{\partial x}$) and northward ($\frac{\partial \phi}{\partial y}$) direction, along with a value for the intercept for the line ($\phi_o$), providing the mean meso-scale value for $\phi$. Formulating this least-squares problem for an overdetermined system of $k$ points as,

$$\min_{x} \|Ax - b\|_2, \tag{3}$$

where $A = \begin{pmatrix} 1 & x_1 & y_1 \\ 1 & x_2 & y_2 \\ ... & ... & ... \\ 1 & x_k & y_k \end{pmatrix}$, $x = \begin{pmatrix} \phi_o \\ \partial_x\phi \\ \partial_y\phi \end{pmatrix}$ and $b = \begin{pmatrix} \phi_1 \\ \phi_2 \\ ... \\ \phi_k \end{pmatrix}$, we solve for $x$ and compute the regression estimates as,

$$x = A^+ b, \tag{4}$$

where $A^+$ is the Moore-Penrose pseudo-inverse. This pseudo-inverse is obtained from the components of singular value decomposition (SVD) of $A$. If the SVD of $A$ is written as $U \cdot \Sigma \cdot V^T$, then $A^+$ is estimated from the inverse of the SVD components as $V \cdot \Sigma^+ \cdot U^T$. Here, $U$ and $V$ are unitary matrices, $\Sigma$ is a rectangular diagonal matrix with A's singular values and $\Sigma^+$ is a rectangular diagonal matrix with the reciprocal of A's singular values. We use the `linalg.pinv` function from the numpy Python library (v1.18.3) to calculate $A^+$.

As a sanity check, we tested the Moore-Penrose pseudo-inverse method of least-squares fitting against the ordinary least-squares fitting by Bony and Stevens (2019), and found no difference between the solutions (not shown). The advantage with incorporating SVD in the regression is that it significantly reduces computing time, because of the availability of vectorized functions in the numpy library.

The Level-4 product includes the eastward (zonal) and northward (meridional) gradients of temperature, pressure, specific humidity as well as $u$- and $v$-winds. Derived from these, Level-4 also provides area-averaged meso-scale divergence ($\mathcal{D}$), vorticity ($\zeta$), vertical velocity ($W$) and pressure velocity ($\omega$), following Bony and Stevens (2019). The dataset also provides the standard error of each of these regressed estimates as ancillaries to the corresponding variables, thus establishing an extent of confidence in the calculation of these meso-scale properties.

Derived variables in Level-4 are at the same vertical grid of 10 m spacing as in Level-3, and the number of sondes regressed at every level is provided as a variable (`sondes_regressed`). If at any level, fewer than 6 sondes have data available, the



value for regressed values at that level is set to NaN. This includes data missing due to no data being recorded as well as sondes removed in any of the previous QC steps. Since the number of sondes regressed change at different levels, this causes abrupt, but generally minor fluctuations in integrated products such as pressure velocity and vertical velocity.

All data variables in Level-4 are along the `circle` and `alt` dimension (see Table 9), and individual sounding data are excluded. The list of sonde IDs included in every circle is included as a variable along dimension `sonde_id`, making it easier to retrieve data for the individual soundings in the circle.

Table 9: Table shows the structure for the Level-4 product, outlining the coordinates, variables and their corresponding descriptions, units and dimensions. The ancillary variables (with the prefix 'se_') give the standard error for their corresponding variables indicated by the suffix in the name.

| OBJECT | NAME | DESCRIPTION | UNITS | DIMENSION |
|---|---|---|---|---|
| Coordinates | alt | height obtained by integrating upwards the atmospheric thickness estimated from the hypsometric equation | m | alt |
| | sounding | sonde number | | sounding |
| | circle | circle number | | circle |
| | circle_lon | longitude of fitted circle for all regressed sondes in circle | degree_east | circle |
| | circle_lat | latitude of fitted circle for all regressed sondes in circle | degree_north | circle |
| | circle_time | mean launch time of all sondes in circle | seconds since 2020-01-01 | circle |
| | segment_id | unique segment ID | | circle |
| Variables | platform_id | platform which flew the circle | | circle |
| | flight_altitude | mean altitude of the aircraft during the circle | m | circle |
| | circle_diameter | diameter of fitted circle for all regressed sondes in circle | m | circle |
| | u | mean eastward wind in circle | $\text{m s}^{-1}$ | circle, alt |
| | dudx | zonal gradient of eastward wind | $\text{s}^{-1}$ | circle, alt |
| | dudy | meridional gradient of eastward wind | $\text{s}^{-1}$ | circle, alt |





Table 9: Table shows the structure for the Level-4 product, outlining the coordinates, variables and their corresponding descriptions, units and dimensions. The ancillary variables (with the prefix 'se_') give the standard error for their corresponding variables indicated by the suffix in the name.

| OBJECT | NAME | DESCRIPTION | UNITS | DIMENSION |
|---|---|---|---|---|
| | sonde_id | unique sonde ID | | circle, sounding |
| | v | mean northward wind in circle | $\text{m s}^{-1}$ | circle, alt |
| | dvdx | zonal gradient of northward wind | $\text{s}^{-1}$ | circle, alt |
| | dvdy | meridional gradient of northward wind | $\text{s}^{-1}$ | circle, alt |
| | q | mean specific humidity in circle | $\text{kg kg}^{-1}$ | circle, alt |
| | dqdx | zonal gradient of specific humidity | $\text{kg kg}^{-1}\ \text{m}^{-1}$ | circle, alt |
| | dqdy | meridional gradient of specific humidity | $\text{kg kg}^{-1}\ \text{m}^{-1}$ | circle, alt |
| | ta | mean air temperature in circle | K | circle, alt |
| | dtadx | zonal gradient of temperature | $\text{K m}^{-1}$ | circle, alt |
| | dtady | meridional gradient of temperature | $\text{K m}^{-1}$ | circle, alt |
| | p | mean air pressure in circle | Pa | circle, alt |
| | dpdx | zonal gradient of pressure | $\text{Pa m}^{-1}$ | circle, alt |
| | dpdy | meridional gradient of pressure | $\text{Pa m}^{-1}$ | circle, alt |
| | D | area averaged horizontal mass divergence | $\text{s}^{-1}$ | circle, alt |
| | vor | area averaged horizontal relative vorticity | $\text{s}^{-1}$ | circle, alt |
| | W | area averaged vertical air velocity | $\text{m s}^{-1}$ | circle, alt |
| | se_dudx | | $\text{s}^{-1}$ | circle, alt |
| | se_dudy | | $\text{s}^{-1}$ | circle, alt |
| | se_dvdx | | $\text{s}^{-1}$ | circle, alt |
| | se_dvdy | | $\text{s}^{-1}$ | circle, alt |
| | se_dqdx | | $\text{kg kg}^{-1}\ \text{m}^{-1}$ | circle, alt |
| | se_dqdy | | $\text{kg kg}^{-1}\ \text{m}^{-1}$ | circle, alt |
| | se_dpdx | | $\text{Pa m}^{-1}$ | circle, alt |





Table 9: Table shows the structure for the Level-4 product, outlining the coordinates, variables and their corresponding descriptions, units and dimensions. The ancillary variables (with the prefix 'se_') give the standard error for their corresponding variables indicated by the suffix in the name.

| OBJECT | NAME | DESCRIPTION | UNITS | DIMENSION |
|---|---|---|---|---|
| | se_dpdy | | Pa m$^{-1}$ | circle, alt |
| | se_dtadx | | K m$^{-1}$ | circle, alt |
| | se_dtady | | K m$^{-1}$ | circle, alt |
| | se_D | | s$^{-1}$ | circle, alt |
| | se_vor | | s$^{-1}$ | circle, alt |
| | se_W | | m s$^{-1}$ | circle, alt |
| | omega | area averaged atmospheric pressure velocity | Pa s$^{-1}$ | circle, alt |

## 6 Summary

The EUREC$^4$A field-campaign took place in January-February, 2020 over the North Atlantic trade-wind region. The campaign
employed a multitude of platforms measuring a range of atmospheric and oceanographic variables with the objective of understanding shallow clouds and processes that influence them. A core part of the campaign was the deployment of dropsondes to characterise the thermodynamic and dynamic structure of the atmospheric environment. Here, we present *JOANNE*, the dataset that provides these dropsonde data and additional derived products.

*JOANNE* presents measurements from 1216 dropsondes launched during EUREC$^4$A by the German research aircraft HALO
and the NOAA WP-3D. Dropsondes were primarily released in groups of twelve circumscribing a mesoscale, $223\,\mathrm{km}$ diameter, circle centered near 57.7°W, 13.3°N, which we call the EUREC$^4$A-circle. Eighty-five circle patterns were flown with dropsonde launches, seventy-three being flown by HALO over the EUREC$^4$A-circle along patterns that were not biased toward particular meteorological conditions. In addition, sondes were launched on circular flight patterns centered elsewhere, along lawn-mower flight patterns coinciding with AXBT drops, and in a variety of other locations to provide context, or calibration for other
measurements. Data presented in *JOANNE* have been quality controlled to eliminate sondes with no, or partially corrupted data. 51 of the 1216 sondes did not provide usable data, and another 98 provided only partial data and are not included in data products from Level-2 onwards.

A comparison of the HALO dropsondes with radiosondes intensively launched from the R/V Meteor close to the western (upwind) edge of the EUREC$^4$A-circle, and with radiosondes launched from the downwind Barbados Cloud Observatory,
suggest a dry-bias. Multiplying relative humidity values by 1.06 appears to largely correct the bias, however due to a lack of

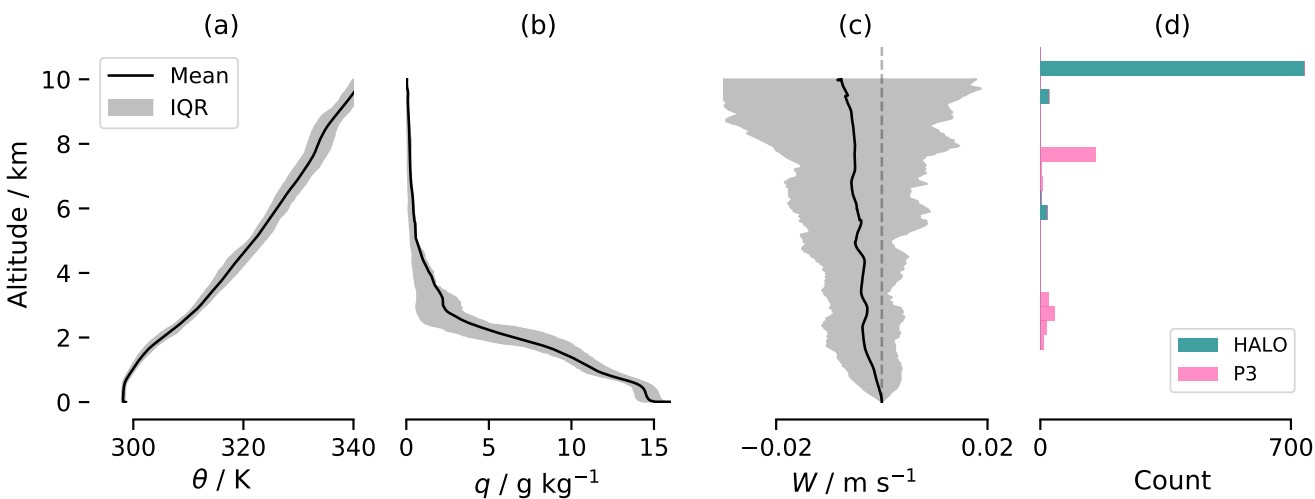

**Figure 6.** Vertical profiles of mean potential temperature (a) and specific humidity (b) from measurements of HALO and P3 sondes. (c) shows vertical profile of mean vertical velocity from estimates of HALO's EUREC[4]A-circle measurements. Shaded regions in (a)-(c) show interquartile range (IQR) of the respective properties. (d) shows the histogram for the flight altitude of dropsondes launched from both platforms.

physical justification, this correction is not applied to the *JOANNE* data. We found no evidence of such a bias in the P3 sondes, and the difference seems attributable to different reconditioning procedures applied on the P3 as compared to HALO.

*JOANNE* is divided in 5 levels of data products, with increasing order of processing and product retrieval. Level-0 comprises the raw measurement data from the dropsondes collected by AVAPS on the aircraft. Level-1 provides data processed using
ASPEN – a state-of-the-art tool for processing raw dropsonde data files. Level-2 consists of individual sounding files that passed through the QC check, but with redundant quantities removed and no derived variables added. Level-3 provides the data after gridding them to a uniform vertical spacing of $10\,\mathrm{m}$, along with derived variables such as potential temperature and specific humidity. Level-4 contains the circle products which are area-averaged, meso-scale variables such as gradients, divergence, vorticity and vertical velocity. The vertical profiles and histogram of flight altitude for dropsonde launches shown in
Fig. 6 provide an overview for a subset of the atmospheric observations that *JOANNE* provides. While reaffirming the typical steadiness in the thermodynamic structure of the trades, *JOANNE* also confirms the high variability in meso-scale vertical motion found by Bony and Stevens (2019) compared to the mean over longer time-scales.

## 7  Code and data availability

The *JOANNE* dataset (George et al., 2021) described in this manuscript is freely available at AERIS (https://doi.org/10.
25326/221). The software used to process the dropsonde data and create *JOANNE* is also publicly available (George, 2021, https://doi.org/10.5281/zenodo.4746313)



*Author contributions.* JOANNE was conceived of by GG. BS and SB designed the sounding strategy for EUREC[4]A, RP and CF adapted this for the P3's participation through ATOMIC. HS and TK contributed to the design and processing of the data. GG and BS performed the quality control. The manuscript was mainly written by GG with contributions by BS. GG, HS, MK, HK, MP and JR were responsible for
dropsonde launch operations and real-time data quality control over different HALO flights. QK and AL were responsible for processing and quality-controlling the data for the P3 flights. All authors read and approved of the manuscript.

*Competing interests.* The authors declare no competing interests

*Acknowledgements.* This research was made possible through generous public support provided to, and managed by, the Max Planck Society (DE), DFG HALO SPP 1294 (DE), CNRS (FR) and NOAA (USA). The project also received funding from the European Research Council
(ERC) under the European Union's Horizon 2020 research and innovation programme (EUREC4A Advanced Grant NO. 694768) and from the French AERIS research infrastructure. A great number of people contributed to the launching of the sondes, as recognized in the EUREC[4]A overview paper. In particular, the authors thank Friedhelm Jansen for his organization and provision of the sondes, Mario Mech and Lutz Hirsch for logistical and technical support, and Angela Gruber for administrative support. Aboard HALO, in addition to the authors, Felix Ament, Jude Charles, Tim Cronin, André Ehrlich, Kerry Emanuel, Florian Ewald, David Farrell, Marvin Forde, Silke Groß, Martin
Hagen, Marek Jacob, Theresa Mieslinger, Ann Kristin Naumann, Theresa Lang, Veronica Pörtge, Sabrina Schnitt, Eleni Tetoni, Ludovic Touze-Peiffer, Jessica Vial, Raphaela Vogel, Antone Wiltshire, Allison Wing and Kevin Wolf contributed to the launching of sondes. Akshar Patel launched the sondes from the P3. The authors are also indebted to the ground and flight crew of both aircraft as well as the civil aviation facility and air traffic control for their efforts to facilitate the measurements. Special thanks to Holger Vömel for advice on the possible cause behind the HALO dry bias. GG thanks Anna Lea Albright, Florent Beucher, Xuanyu Chen, Thibaut Dauhut, Geiske de Groot and
Louise Nuijens in addition to some names already mentioned for feedback on early versions of the dataset and Ann Kristin Naumann for her comments on the manuscript.





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
