# Peer review of "JOANNE: Joint dropsonde Observations of the Atmosphere in tropical North atlaNtic meso-scale Environments"

_Earth System Science Data, 2021_

## Author Comment (AC1)

**Response to reviewers' comments**

JOANNE : Joint dropsonde Observations of the Atmosphere in tropical North atlaNtic meso-scale Environments
doi: https://doi.org/10.5194/essd-2021-162 (https://doi.org/10.5194/essd-2021-162)

**Overarching remarks**

We thank the reviewers for their comments and suggestions on improving the manuscript. We have made changes accordingly and detailed responses to the reviewers' comments are listed below.

One section of the manuscript that both reviewers took interest in was the dry bias in the HALO sondes. From personal communication with Holger Vömel (NCAR), we are now confident that this is the closest we can come to a correction for the dry bias given the information we have. We also received requests from within the community about providing a "moisture-corrected" version of the JOANNE dataset. Therefore, to minimise the effort of users in data processing, especially for variables such as omega, we have now applied the correction to the JOANNE dataset. The correction is made by multiplying the `rh` variable in Level-2 by 1.06. Since all subsequent products of JOANNE are derived from Level-2, the correction automatically applies to variables derived from moisture, e.g. `q` in Level-3 and `omega` in Level-4.

Anticipating that there would be negligible interest in using the uncorrected moisture data and to avoid any confusion among moisture variables, we decided to not include the uncorrected values at all after Level-2. Even so, if there is a need to access uncorrected values, these are still available in Level-0 and Level-1.

The following changes are made in the manuscript in the section describing the HALO dry bias (Section 4):

**"... It is not, however, understood how the contamination of the sensor leads to the bias, and why the multiplicative correction appears to work so well. However, given the information we have, the multiplicative correction is the best option to correct for the dry bias. Therefore, products from Level-2 onwards include this correction. The values in `rh` variable only for the HALO sondes are therefore multiplied by 1.06 when including them in Level-1 from Level-2. All subsequent products in Levels 3 and 4, therefore, have variables derived from the corrected moisture.**

**Users of the data may note that the correction in HALO dropsonde measurements will propagate into other variables, such as PW, pressure velocity and moisture gradients (since it is apparently multiplicative) and will even have a slight effect on estimates of geopotential altitude which depend on the atmospheric density, and hence moisture content. Should the user wish to access the uncorrected moisture values, they can be accessed in Levels 0 and 1."**

A statement in the summary also changes to:

**"Multiplying relative humidity values by 1.06 appears to largely correct the bias, and therefore has been applied from Level-2 onwards to relative humidity and variables derived from it."**

The dataset now has the following changes and carries the version number, v2.0.0.

- Correction of moisture and derived variables from Level-2 onwards

- Addition of the `ugly` flag for `ld_test`, to account for some missing logs in raw data due to which the launch detection test is not carried out (see response to reviewer-2's comments regarding number of P3 sondes in Table-6)

Additionally, we felt that two small changes in the manuscript were needed to make it easier for users who are not familiar with the EUREC$^4$A data environment. Therefore, we made the following changes:

(a) Added the following sentence at the end of the first paragraph of Section 2.2 :
**"More details about HALO's and P3's participation in EUREC$^4$A are provided by Konow et al (2021) and Pincus et al (2021), respectively."**

(b) Added the following in the beginning of Section 5.3.1 :
**"To aid in processing EUREC$^4$A data from aircraft, flight tracks for HALO and P3 were "segmented" into different standard categories such as circles and cloud modules. The flight phase segmentation (FPS) is described in more detail in Konow et al (2021). We use these FPS files to identify the circles and the dropsondes corresponding to these circle segments. To facilitate ..."**

Apart from these additional changes, we have now provided detailed responses to the reviewers' comments below. The boxed text indicates a comment from the reviewer, and the following text is our response to the comment. Changes in the manuscript are quoted in bold.

**Reviewer-1**

**General comments:**

> This manuscript focuses on 1216 dropsondes, which were deployed during the EUREC4A field campaign from two aircraft. These dropsondes were launched in consistent flight patterns, in particular a ~220 km diameter circular flight pattern, to yield estimates of mesoscale vertical motion and generate an unbiased statistical ensemble that characterizes the dynamic and thermodynamic environment of the trades. This dataset can then be used, together with co-located measurements of clouds, to better understand the large-scale environmental controls on cloudiness. This paper is well-written and clearly and concisely describes the different data and data products in JOANNE. The dataset is itself novel and produced with high quality. The acronym is also very clever and a nice tribute to a pioneer in this field. I would recommend publication with minor revisions.

**Specific comments:**

> *Regarding the HALO dropsondes' dry bias, could other platforms from the campaign be included as well? In Sec. 4 and Fig. 5, the authors nicely demonstrate how a multiplicative correction of 1.06 can bring HALO dropsonde measurements in line with radiosonde measurements from the BCO and R/V Meteor, as well as Ron Brown radiosondes (not shown). They also provide a hypothesis for this offset (improper reconditioning and trace pollutants remaining on the humidity sensor). The ATR aircraft and remotely-piloted aircraft RAAVEN were, for instance, also measuring humidity in the vicinity, and could provide further confidence in this 1.06 correction factor.*

The reviewer makes a good suggestion. In fact, while investigating the dry bias problem, we had made comparisons with the ATR and RAAVEN measurements. With the ATR's WVSS2 sensor (which is least affected by cloud droplets), a quasi-systematic difference was observed with

HALO humidity values being roughly 5% lower in terms of RH's absolute percentage points. With the RAAVEN, there were some suspected bias in the measurements, and therefore we discontinued the comparison with HALO dropsondes. However, here in the study, we refrain from making a direct comparison with the HALO dropsonde measurements because of a difference in instrumentation and subsequent processing. The radiosondes and dropsondes use the exact same sensor for humidity (as does RAAVEN) and the differences in post-processing are minimal.

> *More generally, the authors could consider adding an 'uncertainty quantification' section that summarizes various sources of uncertainty in these data. This section could include: uncertainty related to measurement error (e.g. the 'repeatability' of dropsondes, as described in Table 1), which is expected to act like random error; the systematic offset (in specific humidity) of the HALO dropsondes; and the standard errors of regression products (divergence, vertical velocity). This information is already presented in the paper but could be grouped in a short section.*

We agree with the reviewer's idea of summarising the different sources of uncertainty. However, a separate section would come at the cost of coherence in the manuscript, e.g. the dry bias needs to be presented in the context of its evidence and explanation. Nevertheless, we added the following sentences in the summary and linked the relevant parts in the manuscript therein:

**"Possible sources of uncertainty in JOANNE include sensors' repeatability of measurements (see Table-1), uncertainty from the correction of dry bias in HALO sondes (see section-4) and errors arising from the regression estimates (see section-5.3.2)."**

> *Fig. 6 — why only include the IQR? I think it would be useful to also include the 5-95% range, for instance, to better see variability in moisture. Given that cloudiness is relatively low in the trades, moisture variations will mostly be seen in the tails of this distribution, and the IQR does not represent the full picture of moisture variability. I suspect that for temperature, the difference between IQR and a broader interval will be less noticeable.*

We have changed Fig. 6 and its caption accordingly to include 5-95% variability along with IQR.

[Figure]

Revised caption: **"Vertical profiles of mean potential temperature (a) and specific humidity (b) from measurements of HALO and P3 sondes. (c) shows vertical profile of mean vertical velocity from estimates of HALO's EUREC[4]A-circle measurements. Darkly and lightly shaded regions in (a)-(c) show inter-quartile range (IQR) and 5-95 percentile range, respectively. (d) shows the histogram for the flight altitude of dropsondes launched from both platforms."**

> *Could the authors include a bit more background of in situ observations of mesoscale dynamic and/or thermodynamic variability in the trades? Are there previous (smaller) dropsonde datasets for the trades? In the summary, the authors could potentially elaborate on some of the possible scientific applications of these data and data products, though it is not required for a data paper, and this is already a very nice piece of work.*

We have included the following statement in the introduction:

**"Recent instances of dropsonde datasets from tropical field campaigns include ones by Konow et al. (2019) in the northAtlantic summertime tropics and by Vömel et al. (2021) in the tropical east Pacific and Carribean. Konow et al. (2019) published dropsonde data from the second phase of Next Generation Remote Sensing for Validation Studies (NARVAL2). They providedata on a uniform vertical grid at 30 m. JOANNE builds on this idea of a uniform vertical grid from Konow et al. (2019), albeit at 10 m spacing. We go further to provide derived quantities from the circle measurements as well as raw measurement files."**

And we included the following in the summary:

**"JOANNE's immediate usefulness lies in aiding the calibration of or processing the data from remote-sensing instruments onboard HALO as well as for creation of derived products, e.g. a dataset of radiative profiles from EUREC4A soundings (Albright et al, 2021). Furthermore, the dataset potentially has applications in furthering the understanding of processes in the trades, e.g. the influence of mesoscale circulation on clouds (George et al, 2021) or the changes in atmospheric properties within a cold pool (Touze-Peiffer et al, 2021)."**

**Technical corrections:**

> The epigraph from Joanne Simpson at the start of the manuscript is indeed prescient and fits perfectly!

> In the abstract, circle diameter is given as ~222 km, 223 km in summary. Suggest to choose one consistent notation.

In the summary, this has been corrected to say ~222 km.

> Similarly, suggest a consistent notation for the number of circle-means (85 vs. eight-five)

The summary now says "85 circle patterns …"

> In the matrix (A) in Eq. 3 and 4 in the regression, perhaps replace x1 and y1 with delta_x and delta_y for clarity that these are the distances from the circle center to the dropsondes.

The suggested change has been made.

**Reviewer-2**

**General Comments:**

The manuscript presents an extensive data set of 1216 dropsondes launched from the research aircraft HALO and WP-3D during the EUREC4A field campaign in Barbados 2020. The measurement pattern and evaluation method is a consistent evolution of the concept which was tested during the NARVAL2 mission in 2016 and described by Bony & Stevens (2019). The regular drop pattern along circles with a diameter of around 222km allows to derive the mesoscale motion of airmasses in the trade wind region as described previously. In combination with the measured temperature and humidity profiles, this dataset can add crucial information to satellite measurements when assessing the structure and evolution of low clouds in the trades. The evaluation method is comprehensible and the intermediate product levels also allow for a wider use of the data by the scientific community. The paper is well written and provides a comprehensive description of the data and processing steps. Thus, I would recommend publication after minor comments are addressed.

**Specific Comments:**

Fig. 4: The abundance of RH measurements is always lower than the other PTU parameters (p and T). What is the reason for that? Additionally, the RH abundance for the HALO dropsondes in Fig. 4 (a) seems to be lower than for the P3 sondes in (b). Is that difference significant and if so, could that be a hint to problems in the HALO measurement procedure?

Indeed, the abundance of RH measurements is lower than pressure and temperature measurements. This is primarily due to the humidity sensor's longer equilibriation time to ambient conditions compared to that of the pressure and temperature sensors. Typically, the RH sensor starts giving ambiently equilibriated values around 25-30 s after the p & T sensors do. The pre-equilibriation values are discarded by ASPEN QC. This causes fewer data points of RH compared to those of p & T in Level-1 data. We have now clarified this (see end of this response) in the manuscript at the end of the subsection explaining the profile fullness test (Section 3.2).

For the difference between HALO and P3, we traced this back to two primary check failures during the ASPEN processing state. For HALO, there were considerably higher failures in "Filter Check" and "Final Smoothing" for RH. These checks are part of the QC algorithm of ASPEN. The former essentially removes data points that deviate by a certain value after the data series is passed through a low-pass filter. Since we used the standard editsonde configuration, the deviation and the filter wavelength used is 20% and 20 s, respectively. For the final smoothing, all data (p, T, RH and winds) undergo bspline smoothing with a wavelength of 10 s. Therefore, the lower abundance might be due to more noise in the data which causes the ASPEN QC to filter it out. We agree that these differences could serve as a hint that HALO's reconditioning or the lack thereof likely made for noisier data from the humidity sensor, but there is no robust physical explanation backing this hypothesis because how exactly the trace gas pollutants affect the humidity sensor is not yet well-understood.

We have added a statement of information in the manuscript mentioning the two check failures for RH in the ASPEN processing (see below), but we refrain from making further interpretations so as not to introduce speculation in the manuscript.

**"Fig. 4 shows that the abundance of RH values compared to those of pressure (p) and temperature (T) is lower. This is because the RH sensor takes longer to equilibrate to ambient conditions when compared with the other two. This results in fewer measurement records for RH than p and T for the same number of timestamps in the profile.**

**For HALO's RH measurements, there were more failures in ASPEN's *Filter Check* and *Final Smoothing*, compared to those for p and T. These checks are part of the post-processing algorithm of ASPEN. The former removes suspect data that deviate by a certain value after the data series is passed through a low-pass filter. As per the standard editsonde configuration that we use, the deviation and the filter wavelength used is 20% and 20 s, respectively. For the final smoothing, all data (p, T, RH and winds) undergo bspline smoothing with a wavelength of 10 s. (Martin and Suhr, 2021)"**

> Sec 3.3: What is the reasoning behind the specific QC limits assumed (especially Temperature and RH)? Is it based on e.g. ground measurement statistics or models or experience from previous dropsonde measurements?

The near-surface QC bounds that we specified came out not only of previous understanding of the region's wintertime conditions from measurements at the BCO as well as a previous campaign's (NARVAL; Dec 2013) dropsonde measurements, but also keeping in mind the conditions sampled during EUREC4A. Radiosondes from and surface measurements aboard ships and the BCO helped set the bounds. An understanding from previous campaigns of the relative homogeneity in dropsonde measurements spread over areas equivalent to EUREC4A's circle region makes us confident that the bounds chosen are safe enough to allow for small-scale differences, but simultaneously check for faulty data points.

> Table 6: Check for consistency:

We thank the reviewer for bringing these to our notice.

> - # of HALO sondes = 895 compared to 896 stated in sec 2.2

The number stated in sec 2.2 is incorrect. It has been changed to 895, and the corresponding total also to 1215. The same change is also reflected in the abstract and the summary.

> - # of P3 sondes: sec 2.2 states 320 sondes, the ld_test column 316 sondes, the other test columns 322 sondes

We found out that this inconsistency happened due to six missing files (A-type) in the raw data itself, which caused the ld_test to fall into neither *good* or *bad* category. We have now changed the ld_test to also include the *ugly* category in order to account for sondes which cannot be tested for automatic launch detect using the A-files. This has now been corrected and the table now looks like:

| Platform | Classification | ld_test | sat_test | low_test | qc_flag |
|---|---|---|---|---|---|
| HALO | good | 854 | 814 | 831 | 810 |
|  | bad | 41 | 1 | 40 | 41 |
|  | ugly | 0 | 80 | 24 | 44 |
| P3 | good | 310 | 269 | 280 | 263 |
|  | bad | 4 | 0 | 8 | 4 |
|  | ugly | 6 | 51 | 32 | 53 |

In light of the correction above, we have removed the first sentence of the last paragraph in the section describing the launch detection test (Section 3.1) and written the following at the end of the paragraph.

**"A sounding's success in this test is marked by the parameter *ld_test* and takes values of *good* or *bad*, if the corresponding sondes have a successful launch detection or a failed launch detection, respectively. For 6 sondes, A-files were found to be missing in the raw data. These sondes have been tagged as *ugly* for the *ld_test*."**

> - P3 qc_flag: how can 10 sondes be classified as bad? From table 5, it could either be a bad ld_test resulting in 4 bad sondes or a combination of bad sat and bad low test. Since there was no sonde with a bad sat_test, this criterion should produce no bad qc_flag.

See answer to previous comment. There are now only 4 bad sondes for P3, resulting from the 4 failures in launch detection.

> Sec 4: Could you specify how the reconditioning procedure differs between the two aircraft? I would assume the initialization procedure should be very similar?
> Could the "measurement history" of the HALO sondes also be a reason for the dry bias? Compared to the radiosondes and the P3 sondes, the HALO sondes all came down from the cold and rather dry upper troposphere. Maybe one could compare HALO soundings with lower drop altitudes if available?

The initialization procedure for the RD-41 dropsondes is the same for both aircraft. However, reconditioning operations are to be carried before operations start, ideally before the flight starts. The procedure is similar to the initialization, with the exception that instead of readying the sonde for measurements, hardware configuration tests are run to "remove pollutants from RH sensor". The reconditioned sonde stores the information about whether this test was performed and if it was successful. All sondes have to be reconditioned before any initialization because the hardware tests cannot be run during the active data-collection phase.

We have made the following changes in the dry bias section to clarify this point:

**"The most plausible explanation is the lack of reconditioning for HALO dropsondes, which resulted in some trace gas pollutants being retained on the humidity sensor and should otherwise have been removed during the reconditioning. The P3 sondes were reconditioned before data collection and the protocols for P3 and HALO therefore differed in this aspect. This leads us to believe that the dry bias is observed only in HALO dropsondes because of the absent reconditioning."**

The statement in the summary has also been made clearer:

**"We found no evidence of such a bias in the P3 sondes, and the reason for the dry bias in HALO seems attributable to a lack of reconditioning of the HALO sondes."**

The "measurement history" hypothesis was actually the reason for a dry bias observed in the previous version of dropsondes, RD-92. However, for RD-41 (EUREC[4]A sondes), the RH sensor has been improved to include an in-built sensor-heating mechanism which essentially removes any influence of "history" and therefore should be unaffected by flight altitude or previous conditions in its path.

> 5 and text: You state that the multiplicative correction works well for all heights. However, there is some discrepancy in Fig.5 (c) where HALO and Meteor pdfs align pretty well (even without dry bias correction) and the BCO soundings which seem to be moister. Please comment or modify in the text.

The RH distribution in Fig. 5c is for altitudes above the inversion layer, i.e. in the free troposphere. The lower moisture values here therefore don't seem to show any substantial effect due to the correction, compared to those seen at lower altitudes – confirming that the bias is indeed multiplicative. This is what we meant by "successful at all heights".

Indeed, the BCO soundings seem to be moister. At this point, we cannot explain why that is. Since they are also moister than the Meteor soundings, it is likely a physical signal. Our reasoning for the dry bias correction here is that for the moist area, i.e. surface to inversion

height (~2 km), it works seemingly well. For the drier upper layers, the multiplicative correction does not make a subtantial difference to the measurements owing to the low magnitudes.

> Sec 5.2.1:
>
> - To 4.: Could you explain the reason behind this procedure? Do you intend to get rid of short scale (adiabatic) vertical motion effects in the profile?

Yes, since specific humidity and potential temperature are conserved variables, interpolating them linearly in physical space will avoid errors in interpolation that might arise from RH and temperature, although this causes negligible differences.

> - To 5.: Do you first calculate u and v from Level2 data (windspeed and direction) for the interpolation and then recalculate windspeed and direction?

Yes. Point-1 in Section 5.2.1 mentions that we calculate $u$ and $v$ from Level-2, and point-5 mentions that we then recompute the windspeed and wind direction from the interpolated $u$ and $v$ values.

**Technical Corrections:**

> l 203: less instead of lesser?

Corrected.

> l 300: ) missing

Corrected.